# Unpacking lithic assemblage variability in the Early Upper Palaeolithic: A multivariate approach to the structure of the Iberian Aurignacian

**Timothy Canessa**[1,2]*, **Paloma de la Peña**[3,4]

1 Department of Prehistoric and Historical Archaeology, University of Vienna, Vienna, Austria, 2 Human Evolution and Archaeological Sciences (HEAS), University of Vienna, Vienna, Austria, 3 Departamento de Prehistoria y Arqueología, Universidad de Granada, Campus Universitario de Cartuja, Granada, Spain, 4 Evolutionary Studies Institute, University of the Witwatersrand, Johannesburg, South Africa

* timothy.canessa@univie.ac.at

## Abstract

The Aurignacian technocomplex of the Early Upper Palaeolithic remains a long-standing focal point for understanding the expansion of modern humans across Europe. Diagnostic assemblages occur across vast swathes of the continent, suggesting the existence of broadly connected groups and traditions around 43–32 ka cal BP. However, while its extensive distribution is often regarded as proxy evidence for the spread of modern human groups, artefact assemblages are known to be synchronically and diachronically variable in ways that reveal an inconsistent representation of diagnostic traits. In the Iberian Peninsula, this variability is exemplified by an idiosyncratic material record in which diverse Aurignacian assemblages occur alongside undiagnostic or 'culturally indeterminate' ones, leading many Aurignacian occupations to be disputed. In this paper, we assimilate this regional record through quantitative analyses of techno-typological attributes from all sufficiently published and chronologically relevant assemblages of the Early Upper Palaeolithic. Using two multivariate techniques, we first explore associations between assemblages and thereafter test whether inter-assemblage variability is related to spatial and temporal distances. Our results cast light on the spatial structure of variability by revealing that inter-assemblage differences increase with spatial distance but show no linear relationship to temporal distance. This spatial finding challenges the cross-regional applicability of the Aquitaine model of techno-typological change, whilst the absence of temporally structured variability suggests a heterogeneous representation of diagnostic traits across and within temporal classes of assemblages.

## Introduction

Recent archaeological and genetic evidence has shown that Initial Upper Palaeolithic assemblages from Bacho Kiro (Bulgaria) and Ilsenhöle (Germany) were the product

**Data availability statement:** All relevant data and code are provided in the associated Zenodo repository (https://doi.org/10.5281/zenodo.18462036) and Supporting Information files of the paper.

**Funding:** This work was supported by a Gibraltar Government Scholarship and John Mackintosh Trust Grant awarded to TC and a Ramón y Cajal research contract (RYC2020-029506-I) awarded to PdIP. The funding bodies had no role in the study design, data collection and analysis, decision to publish, or preparation of the manuscript.

**Competing interests:** The authors have declared that no competing interests exist.

of early *Homo sapiens* dispersals around 45 ka cal BP [1,2]. However, available evidence also suggests that these early dispersals constituted small, potentially isolated populations that left no genetic trace in later hunter-gatherer groups [2–4]. Therefore, while the first European appearance of modern humans may be linked to Initial Upper Palaeolithic technocomplexes, current data continues to support a more significant expansion with the Aurignacian of the Early Upper Palaeolithic c. 43–32 ka cal BP [5–9]. In this context – and due also to its novel techno-cultural elements – the Aurignacian continues to be a focal point for examining the consequential emergence and behaviours of groups across the many regions where it occurs.

Since its initial identification over 150 years ago at Grotte d'Aurignac [10], the Aurignacian has passed through various cycles of description and definition by prehistorians [11–20]. One noteworthy contribution from more recent years concerns the consolidation of the technocomplex's structure into four chrono-cultural phases (i.e., Proto-Aurignacian, Early Aurignacian, Evolved Aurignacian, Late Aurignacian) based on the application of renewed chronometric and technological data to the classic sequences of the Aquitaine Basin (southwest France) [e.g., 21–28]. On a technological level, the first two phases are often distinguished by significant differences in the production of blades and bladelets: in the Proto-Aurignacian, both blades and bladelets are attributed to the sequential reduction of a single core [21,22] whereas in the Early Aurignacian, blade and bladelet production derive from two independent systems of reduction, with blades obtained from prismatic cores and bladelets obtained from carinated end-scraper cores [21,29]. However, recent work has shown that these discrete differences are not universally demonstrated [30–34], and there is additional uncertainty over whether the two variants constitute chronologically successive phases [25,35] or synchronous expressions of an overarching industry [21,26,30,34,36,37].

Notwithstanding this, the chronology and stratigraphic position of the Proto-Aurignacian and Early Aurignacian have long made them central to discussions on the expansion of modern humans and disappearance of Neanderthals [e.g., 6,7,38–42]. Following Mellars [43,44], the prevailing model of this expansion proposes a westward dispersal from the Levant via a southern and northern route (i.e., Mediterranean and Danubian) for the Proto-Aurignacian and Early Aurignacian respectively. Current chronological and archaeological data generally retain this as the dominant model, although the origin of the Early Aurignacian remains a particularly elusive matter given that Central European assemblages predate analogous Levantine ones [36,45]. Similarly, other researchers have questioned the ancestral relationship between the Proto-Aurignacian and the Early Ahmarian of the Levant due to technological and chronological discrepancies [46,47]. Nevertheless, although further work is needed to clarify these points, Mellar's [43,44] model and competing ideas of its origin, evolution and spread [e.g., 12, 15, 40, 48–50] continue to provide the main theoretical context for evaluating the chrono-cultural variability of the Aurignacian *sensu lato* – an aspect of the technocomplex that has been repeatedly highlighted [e.g., 51–59].

Indeed, a significant consequence of this observed variability is that, rather than being a unified "package" of discrete behavioural traits, the Aurignacian and its

chrono-cultural variants are diversely represented across space and time [52]. Various regional syntheses and studies of lithic assemblages have repeatedly proven this point [31,33,57,60], substantiating previous descriptions of the Aurignacian as a mosaic rather than monolithic phenomenon [61]. In this context, growing evidence has come to challenge the application of the paradigmatic Aquitaine model to other regional records [34, 31, but see also 62] and raise important questions about the technocomplex's footing in the Early Upper Palaeolithic. For example, how do we reconcile this variability with the notion that the Aurignacian is a strongly shared tradition? And, what role do culturally indeterminate assemblages play in our understanding of the Aurignacian and Early Upper Palaeolithic at large?

Hence, in this paper, we address some of these questions and themes with a focus on the Early Upper Palaeolithic record of the Iberian Peninsula, a region that represents one of the most interesting case studies of Aurignacian assemblage variability. The overarching objectives of our study are to elucidate the extent of lithic assemblage variability and assess its spatio-temporal dimension. In doing so, we aim to generate new insights into the structure of the regional Aurignacian and Early Upper Palaeolithic more broadly.

## The Early Upper Palaeolithic of Iberia

As proxy evidence for the spread of modern human groups, sites with diagnostic Aurignacian assemblages are found across the entire continent. However, their uneven distribution displays a diverse signal of settlement possibly shaped by climate [63,64] and the distribution and stability of social networks [65]. Noteworthy clusters of sites in Central Europe, south-west France and northern Iberia contrast strongly with more scattered distributions in parts of Central and Eastern Europe, the Balkans, Britain and the Italian and Iberian peninsulas, which may or may not reflect occupation intensity. As is often the case, research and sampling histories likely play a significant role in producing this pattern [see 66].

At the south-west margin of this broad spatial setting, the Iberian Peninsula is itself host to a variable record of human occupation and behaviour. Most glaringly, the greater density of sites in the north is a strong divergence from the rest of the peninsula, where the extent of human occupation is scanter, particularly in the interior and southern regions. Only recently (i.e., the last 10 years) has this faint signal been slightly amplified by the excavation of new and previously explored sites [e.g., 67–73], in turn demonstrating the well-known effect of research bias on archaeological data. Yet despite this recent work, current evidence still paints an intriguing picture of human settlement that remains heavily debated. In northern Iberia, a relatively early chronology for the Aurignacian is indicated by Proto- and Early Aurignacian occupations at Labeko Koba (Basque Country), La Viña (Asturias) and L'Arbreda (Catalonia), which collectively date to the 42–40 ka cal BP interval [74]. In addition, an even earlier appearance has been proposed for the so-called 'Transitional Aurignacian' of El Castillo (Cantabria) [75] which recent Bayesian modelling of old and new dates suggests began c. 45.5 ka cal BP [76]. Likewise, Bayesian modelling of radiocarbon dates obtained with the ultrafiltration protocol indicates a degree of chronological overlap between the Châtelperronian and Aurignacian around 43.5–41.5 ka cal BP in Cantabria, with the Aurignacian predating the Châtelperronian by almost one millennium [77, but see also 78]. In the southern half of the peninsula, the asymmetrical chronology and character of the Aurignacian has long been a topic of frequent debate [e.g., 79–81] and some authors have recently argued that Aurignacian groups reached these areas as early as 43–41 ka cal BP [82, 72, but see 83, 84, 85], in contradiction to claims that the earliest southern arrival occurred with the Evolved Aurignacian c. 37 ka cal BP [67,70,86]. Based on current data, it remains unclear whether a delayed southern arrival relates to a regional persistence of the Middle Palaeolithic [87] and/or possible biogeographical barrier along the Ebro River basin (i.e., the Ebro Frontier model) [85,88].

In terms of the lithic evidence of these occupations, several statements can be made about the variability of assemblages dated to the regional Aurignacian timeframe (i.e., 43–32 ka cal BP). Firstly, at both diachronic and synchronic scales, the character and composition of lithic assemblages is rather diverse, constituting a mixed set of techno-typological features sometimes accompanied by few diagnostic elements, e.g., Aurignacian blades (for the Early

Aurignacian), Dufour bladelets (both Dufour and Roc-de-Combe subtypes), carinated end-scraper cores and bladelets produced from carinated/nosed end-scraper and burin cores. This is particularly true in southern Iberia, where lower densities of material suggest more ephemeral occupations (e.g., Pego do Diabo, Cova de les Malladetes, Finca Doña Martina, La Boja) unlike the richer deposits at many northern sites (e.g., La Viña, Labeko Koba, Cueva Morín) [67,70,89–92]. Secondly, in some cases, diagnostic elements are insufficient or absent to support a secure attribution to the Aurignacian technocomplex. This has led some researchers to classify them as 'indeterminate Early Upper Palaeolithic assemblages' or nothing at all, e.g., Cova Gran level 497D, Gorham's Cave level CHm.5, Abrigo de Sopeña levels VIII–XI, Lapa do Picareiro levels DD and FF and Cova de les Cendres level XVII [69,72,93–95]. Given this variability, the Iberian record does not seem to align neatly with the neighbouring Aquitaine basin and its emblematic, often rich deposits preserving multiple Aurignacian horizons. Yet this misalignment is not exclusive to the Aurignacian but also relevant to the Iberian Gravettian [96], suggesting that among these early expressions of the Upper Palaeolithic, pronounced variability may be the norm rather than exception.

In proposing explanations for the nature of this material record, it is reasonable to suggest that climatic downturns may explain the absence of sedimentary deposits with Early Upper Palaeolithic material in some areas and that, additionally, disturbance by syn- and post-depositional processes may be causally related to the composition of others, particularly (but not exclusively) those from sequences that document a Middle-to-Upper Palaeolithic transition [e.g., 97–101]. For example, evidence of discoidal and Levallois knapping among various assemblages (e.g., Cueva Morín, La Viña, Aitzbitarte III) could be attributed to vertical displacements of material [91,102,103]. Conversely, if geological attrition is not the cause, the underrepresentation of certain artefact types (e.g., bladelets) can be a consequence of the more rudimentary methods used in old excavations (i.e., lack of sieving) [see 104]. However, these proposals must also be evaluated against the less explored question of whether these techno-typological elements accurately represent the diverse spectrum of knapping behaviours and toolkits during the Early Upper Palaeolithic. In this endeavour, both diagnostic and undiagnostic assemblages have an important role to play, for while the latter are rarer, they may nonetheless attest to meaningful behavioural variation. In Iberia, the flake-based assemblage 497D of Cova Gran and its non-Aurignacian character is a typical case in point; despite an age of c. 37 ka cal BP, blades and bladelets are comparatively minimal and the retouched tool component is dominated by side-scrapers, notches and denticulates [93,105].

As others have already argued, much of historical research on the Early Upper Palaeolithic of Iberia has been characterised by the cultural classification of assemblages based on typological criteria, particularly the presence of index fossils [106]. Indeed, the recent redating of Cantabrian sites placed several assemblages previously classified as Aurignacian or Gravettian (on typological grounds) to later Upper Palaeolithic periods [77]. Other publications have assigned taxonomic labels (i.e., Aurignacian or its sub-variants) to lithic assemblages strongly based on the chronologies of relevant deposits, in part due to advancements in radiocarbon dating and sample pretreatment [72,73,82]. However, one consequence of this method of inference and the accompanying imposition of cultural taxonomies is that attention is drawn away from the composition of lithic assemblages and their connection to other realms of explanation. As a result, inter-assemblage similarities/differences and the drivers of this variability are rarely explored and tested, even though this remains a discipline-defining subject of Palaeolithic archaeology [107–111].

In this context, this paper makes use of a large, newly compiled dataset of Early Upper Palaeolithic assemblages (n = 41) to explore their techno-typological composition and quantify the extent of variability across the entire Iberian Peninsula (Table 1 and Fig 1). Using a combination of multivariate statistical techniques, we examine the structure of this variability and test its spatial and temporal dimensions. As such, our quantitative pan-Iberian study presents an original approach to the study of variability in the Iberian Early Upper Palaeolithic, a subject typically investigated with regional and descriptive approaches.

**Table 1. List of analysed sites and assemblages in order of decreasing latitude.**

| Site | Type | Latitude | Longitude | Level | Assemblage ID | Technocomplex | Reference |
|------|------|----------|-----------|-------|---------------|---------------|-----------|
| Cueva Morín | Cave | 43.372 | −3.850 | 8 | MORÍN-8 | Proto-Aurignacian | [102] |
| Cueva Morín | Cave | 43.372 | −3.850 | 9 | MORÍN-9 | Proto-Aurignacian | [102] |
| La Viña | Rock shelter | 43.313 | −5.827 | XII | VIÑA-XII | Early/Evolved Aurignacian | [91] |
| La Viña | Rock shelter | 43.313 | −5.827 | XIII | VIÑA-XIII | Early Aurignacian | [91] |
| La Viña | Rock shelter | 43.313 | −5.827 | XIII(inf.) | VIÑA-XIII(inf) | Proto-Aurignacian | [91] |
| El Castillo | Cave | 43.292 | −3.965 | 16 | CAST-16 | Proto-Aurignacian | [112] |
| El Castillo | Cave | 43.292 | −3.965 | 18B | CAST-18B | Possibly Aurignacian | [113] |
| El Castillo | Cave | 43.292 | −3.965 | 18C | CAST-18C | Possibly Aurignacian | [113] |
| Aitzbitarte III | Cave | 43.263 | −1.895 | Vb-central | AITZB-III-Vb-c | Evolved Aurignacian | [103] |
| Aitzbitarte III | Cave | 43.263 | −1.895 | Vb-base | AITZB-III-Vb-b | Possibly Aurignacian | [103] |
| Ekain | Cave | 43.237 | −2.276 | IXb | EKAIN-IXb | Evolved Aurignacian | [114] |
| Labeko Koba | Cave | 43.056 | −2.488 | III | LKOBA-III | Possibly Aurignacian | [90] |
| Labeko Koba | Cave | 43.056 | −2.488 | IV | LKOBA-IV | Early Aurignacian | [90] |
| Labeko Koba | Cave | 43.056 | −2.488 | V | LKOBA-V | Early Aurignacian | [90] |
| Labeko Koba | Cave | 43.056 | −2.488 | VI | LKOBA-VI | Early Aurignacian | [90] |
| Labeko Koba | Cave | 43.056 | −2.488 | VII | LKOBA-VII | Proto-Aurignacian | [90] |
| L'Arbreda | Cave | 42.161 | 2.746 | H | ARBRED-H | Proto-Aurignacian | [115,116] |
| Cova Gran | Rock shelter | 41.927 | 0.813 | 497D | GRAN-497D | Indeterminate Early Upper Palaeolithic | [117] |
| Cova Foradada (Calafell) | Cave | 41.205 | 1.581 | IIIc | FORC-IIIc | Early Aurignacian | [68] |
| Abrigo de la Malia | Rock shelter | 41.007 | −3.258 | LU-V | MALIA-LU-V | Evolved Aurignacian | [73] |
| Cardina-Salto do Boi | Open air | 40.979 | −7.101 | GFU 5/UA10 | CARD-G5-A10 | Evolved/Late Aurignacian | [71] |
| Lapa do Picareiro | Cave | 39.530 | −8.653 | DD | LAPA-DD | Possibly Aurignacian | [118] |
| Lapa do Picareiro | Cave | 39.530 | −8.653 | FF | LAPA-FF | Possibly Aurignacian | [118] |
| Lapa do Picareiro | Cave | 39.530 | −8.653 | GG | LAPA-GG | Early Aurignacian | [118] |
| Lapa do Picareiro | Cave | 39.530 | −8.653 | HH | LAPA-HH | Early Aurignacian | [118] |
| Lapa do Picareiro | Cave | 39.530 | −8.653 | II | LAPA-II | Early Aurignacian | [118] |
| Gato Preto | Open air | 39.336 | −8.930 | C | PRETO-C | Evolved Aurignacian | [89] |
| Cova de les Malladetes | Cave | 39.021 | −0.300 | XII | MALLAD-XII | Late Aurignacian | [70] |
| Cova de les Malladetes | Cave | 39.021 | −0.300 | XIII | MALLAD-XIII | Evolved Aurignacian | [70] |
| Cova de les Malladetes | Cave | 39.021 | −0.300 | XIVA | MALLAD-XIVA | Evolved Aurignacian | [70] |
| Gruta Pego do Diabo | Cave | 38.863 | −9.220 | 2 | DIABO-2 | Late Aurignacian | [89] |
| Cova Cendres | Cave | 38.686 | 0.155 | XVIC | CEND-XVI-C | Late/Evolved Aurignacian | [69] |
| La Boja | Rock shelter | 38.079 | −1.490 | OH15 | ADB-15 | Late Aurignacian | [67] |
| La Boja | Rock shelter | 38.079 | −1.490 | OH16 | ADB-16 | Late Aurignacian | [67] |
| La Boja | Rock shelter | 38.079 | −1.490 | OH17 | ADB-17 | Evolved Aurignacian | [67] |
| La Boja | Rock shelter | 38.079 | −1.490 | OH18 | ADB-18 | Evolved Aurignacian | [67] |
| La Boja | Rock shelter | 38.079 | −1.490 | OH19 | ADB-19 | Evolved Aurignacian | [67] |
| La Boja | Rock shelter | 38.079 | −1.490 | OH20 | ADB-20 | Evolved Aurignacian | [67] |
| Finca Doña Martina | Rock shelter | 38.079 | −1.490 | 8 | FDM-8 | Evolved Aurignacian | [67] |
| Bajondillo | Cave | 36.623 | −4.497 | 11 | BAJO-11 | Evolved Aurignacian | [82,119] |
| Gorham's Cave | Cave | 36.120 | −5.342 | CHm.5 | GOR-CHm5 | Indeterminate Early Upper Palaeolithic | [94] |

Technocomplex attributions reflect those from corresponding publications with the exception of the 'possibly Aurignacian' category (see 'Materials and methods' section). The references provided also represent the sources of techno-typological data.

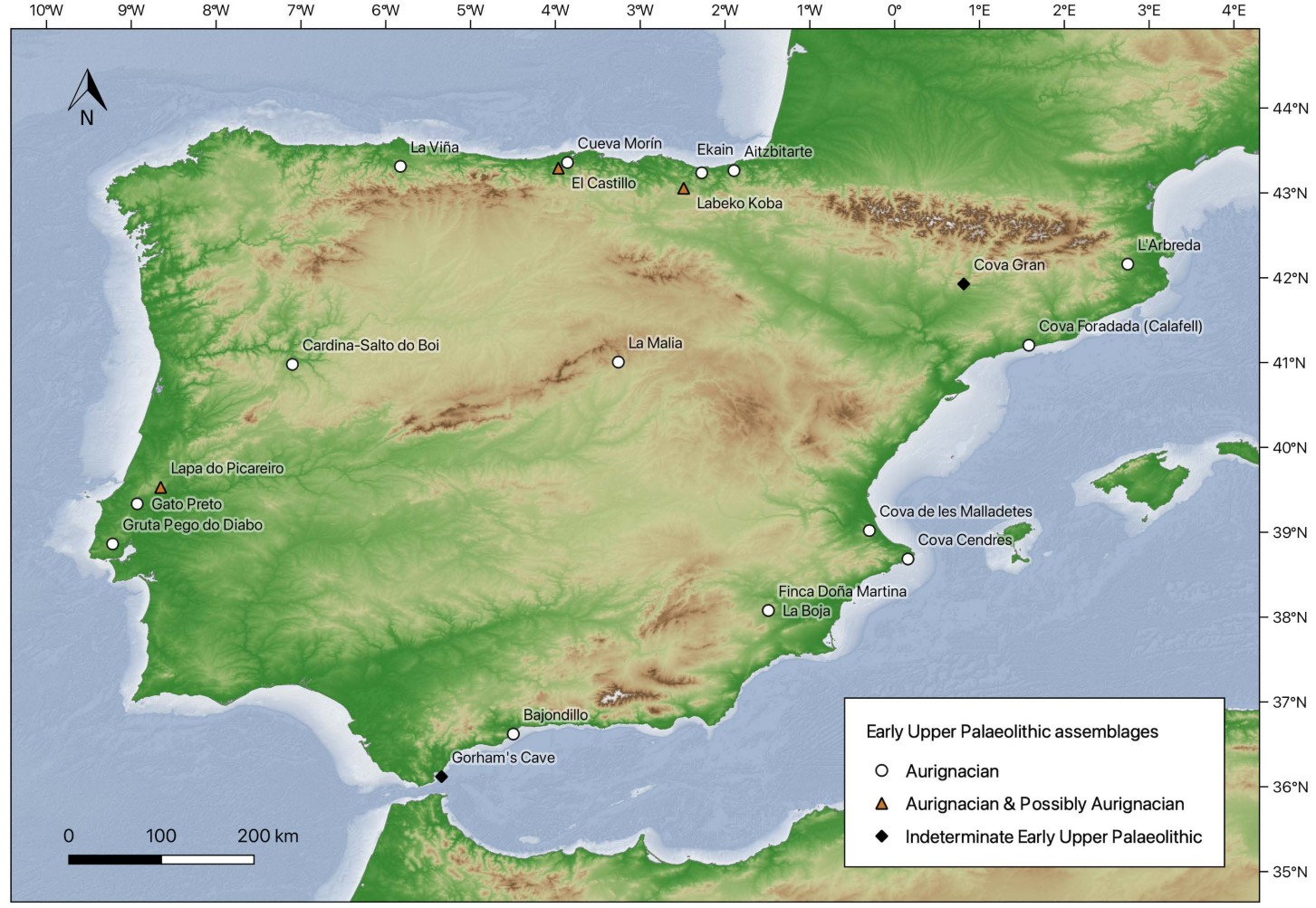

**Fig 1. Map showing the analysed sites.** The Digital Elevation Model (DEM) and ocean bathymetry data were obtained from the USGS EROS Centre (https://www.usgs.gov/centers/eros/data) and EMOD network (https://emodnet.ec.europa.eu). Technocomplex attributions provided in the map legend reflect those given in corresponding publications (where provided). See 'Materials and methods' section for details and Table 1 for the attribution of each assemblage.

## Materials and methods

### Lithic assemblage dataset

A broad dataset of 41 Early Upper Palaeolithic assemblages was compiled following an extensive review of published literature. This dataset records the presence-absence of numerous technological and typological attributes, serving as a basis for assessing inter-assemblage variability (Fig 2 and S1 Table). A binary (i.e., presence-absence) format was utilised to maximise the number of assemblages and therefore the geographic and chronological scope of the study. This format sacrifices a more fine-grained resolution of variability in favour of a larger sample but is nevertheless well suited to multivariate analyses [e.g., 120–122]. As one might expect, the reporting of technological and typological data is too inconsistent to allow for statistical analysis of continuous variables across such a large sample of assemblages. For example, a thorough review of the literature demonstrates that some publications provide artefact counts for typological categories but not technological ones (e.g., flake cores, core tablets, crested pieces) and vice versa. Additionally, in some cases,

**Fig 2. Binary heatmap of techno-typological attributes among the studied assemblages.** Assemblages in the heatmap are ordered by decreasing latitude as per Table 1. Black = presence, grey = absence.

counts for two separate artefact categories are grouped into one (e.g., n=blades-bladelets), preventing any understanding of their relative frequencies. A binary format therefore helps to minimise these problems whilst still enabling the analysis of broad technological and typological attributes, as per the objective of this study.

In selecting assemblages for analysis, several important criteria were used to determine their suitability. Firstly, only dated assemblages from stratified contexts securely attributed to the Early Upper Palaeolithic *sensu lato* were chosen. Four exceptions to this rule are the undated assemblages of Finca de Doña Martina level 8, Lapa do Picareiro level HH, Labeko Koba level III and Aitzbitarte III level Vb-base, as these Early Upper Palaeolithic assemblages either overlie or underlie dated Aurignacian layers [67,72,103,123]. Secondly, only assemblages published with sufficiently detailed data were included in the dataset. One result of these criteria is that various assemblages described as Aurignacian (or Early Upper Palaeolithic) and/or with ages consistent with the Aurignacian timeframe were excluded from the dataset. Some examples include L'Arbreda level G and Cova Beneito levels B8–B9, for which insufficient technological data exists for this study [124–126] and Cardina-Salto do Boi levels GFU5-A9 and GFU5-A8, whose retouched tool inventories are, to our knowledge, not yet fully published. Major reworking of deposits and/or the uncertain provenance of lithic artefacts also precluded the selection of assemblages, as is the case with the Gruta de Salemas (Portugal), Vascas (Portugal), Boquete de Zafarraya (Spain) and Lezetxiki (Basque Country), all of which have been reported to contain diagnostic Aurignacian artefacts [89,127,128].

There are several sites and assemblages which are subjects of debate regarding their attribution to the Aurignacian or the Early Upper Palaeolithic more generally. Most notably, these include El Castillo levels 18B and 18C, described as a 'Transitional Aurignacian' [129: 529] or 'Transitional/Initial Upper Palaeolithic' assemblage [130: 3]; Bajondillo level 13, recently reinterpreted as a 'chronologically early Aurignacian' dated to 43 ka cal BP [82: 210]; and Lapa do Picareiro levels GG–II, an aggregated assemblage documenting an 'early Aurignacian' occupation circa 41–38 ka cal BP [72: 1]. All these interpretations have been critically appraised by researchers who have pointed out individual (but often shared) issues with the associated evidence and argumentation. These primarily centre on the evidence of reworked deposits, the incoherence of radiocarbon dates and the heavy reliance on chronological data for cultural diagnoses [83,81,84,85]. Although all Pleistocene sites are palimpsests shaped by mixing, these arguments raise important questions about their validity as expressions of the Aurignacian.

Factoring in these issues, we have excluded Bajondillo level 13 from the final selection of assemblages based on the absence of stone tool evidence supporting an Aurignacian attribution [see 83, 84] and the possibility of mixing between Middle and Early Upper Palaeolithic deposits [131]. In addition, it should be noted that the recent claim of an 'early Aurignacian' occupation in level 13 is based solely on chronological evidence [82], which provides no reassurances on its cultural attribution. On the other hand, we have cautiously retained El Castillo levels 18B and 18C on the basis that the diagnostic artefacts found within them (e.g., bone tools, Aurignacian blades, Dufour bladelets, carinated and nosed end-scraper cores) warrant the assemblages to be compared with others in our analysis – some of which also contain combined Middle and Upper Palaeolithic elements, e.g., Cueva Morín levels 8 and 9, and La Viña levels XIII and XIII(inf.). In our view, more evidence (particularly geoarchaeological, genetic and proteomic evidence) is required to ascertain the degree of reworking and the authorship of the assemblages, the latter of which may be attributed to Neanderthals if we accept recent metric analyses of three deciduous tooth crowns from level 18B [130]. Lastly, we have also retained Lapa do Picareiro levels GG–II based on the agreement between the chrono-stratigraphic evidence and stone tool characteristics. However, rather than treat levels GG–II as a single aggregated assemblage [72], we include them in our analyses as three separate entities [as per 118].

The cultural attribution of assemblages used in our study (see Table 1) follows those provided in original publications (when given). These include the various sub-variants of the Aurignacian and the term 'indeterminate Early Upper Palaeolithic'. For Gorham's Cave level Chm.5 – an assemblage dated to 36–33 ka cal BP – the term 'indeterminate Early Upper Palaeolithic' is used to rephrase its original description as a 'poorly characterised Initial Upper Palaeolithic' [94: 178]. On

the other hand, the attribution of Cova Gran level 497D reflects the exact wording used in the relevant publication, where the authors considered it 'prudent to assign level 497D to an indeterminate Early Upper Palaeolithic' [93: 218].

For assemblages not given a cultural attribution (i.e., not sufficiently described or classified) but chronologically and/or archaeo-stratigraphically suggestive to be Aurignacian, we have conservatively categorised them as 'possibly Aurignacian'. However, as an additional exception, this category has been applied to El Castillo levels 18B and 18C to avoid the speculative and loaded quality of the 'Transitional Aurignacian' label [129].

**Technological attributes.** Technological and typological attributes recorded for each assemblage encompass a range of general artefact categories pertinent to the Early Upper Palaeolithic and Aurignacian technocomplex (see S1 Fig for a visualisation). The thirteen technological attributes used for analysis include a range of different blank and core technologies, as well as the presence-absence of specific core types, including carinated and nosed end-scraper cores (coded *Carinated.EndscraperCores* and *Nosed.EndscraperCores*) which are commonly regarded as index fossils of the Aurignacian [132]. Burin cores (coded *BurinCores)* are not typically viewed as index fossils but represent distinct approaches to bladelet production that are argued to be characteristic of the technocomplex, at least in some French Aurignacian assemblages [133]. Other generic core types connected to blade and bladelet production are not included in the analysis due to inconsistencies in how they are reported and described – some publications describe cores morphologically (e.g., polyhedral, prismatic) whilst others describe them technologically (e.g., blade cores, bladelet cores), making comparison problematic. While not ideal, *BladeBlanks, BladeletBlanks* and *FlakeBlanks* capture the presence-absence of blank production schemes and are therefore attributes that provide some technological information.

*DiscoidalTech* and *LevalloisTech* are categories that represent blanks and cores specifically described by authors with the terms 'discoidal' or 'Levallois'. Similarly, *BipolarTech* combines blanks and cores described as 'bipolar' (i.e., from percussion on anvil) with the exception that where authors make an explicit distinction between 'splinters' and 'splintered pieces', the former are understood as bipolar blanks, e.g., Cova de les Cendres [69]. We wish to underline that the collection of data has been faithful to how they are reported, irrespective of any disagreement with the interpretation of specific artefacts. If artefacts are not mentioned as being present (beyond their absence in a table with counts, if provided) this is taken as evidence of absence. *FlakeCores* is a category distinct from *DiscoidalTech and LevalloisTech* by virtue of core descriptions that avoid reference to discoidal and Levallois schemes, e.g., "centripetal cores", "prismatic cores for flakes" or simply "flake cores". *ScraperCores* are core-scrapers which are not described as 'carinated' or 'nosed', equivalent to items 15 and 16 of the de Sonneville-Bordes and Perrot [134] type-list when it is used for the reporting of retouched tool data (as is often the case).

*CoreWorking* refers to evidence of core preparation and maintenance and amalgamates core tablets, core flanks, core rejuvenation flakes and crested/semi-crested pieces (S1 Fig). These items derive from the volumetric exploitation of cores for blades and bladelets and, in the case of core tablets and crests, are strong characteristics of Upper Palaeolithic technology only found in a few Middle Palaeolithic contexts [e.g., 135–138]. Likewise, *BurinSpalls* are a by-product of burin production – a technological practice shared with some Initial Upper Palaeolithic contexts [e.g., 139, 140] – and can be technologically indicative when burins are absent in an assemblage.

**Typological attributes.** The recorded typological attributes encompass 18 retouched tool categories that range from generic (e.g., *Ret.Blades*, *Ret.Bladelets* and *Ret.Flakes*) to more discrete tool types (e.g., *BackedBlades*, *BackedBladelets*, *EndScrapers*, *SideScrapers*, *Burins*). Taxonomically relevant tool types of the Aurignacian are also recorded, namely *AurignacianBlades* and Dufour bladelets (both Dufour and Roc-de-Combe subtypes, coded as *DufourDufour* and *DufourRoc.de.Combe*). So-called Aurignacian blades were commonly regarded as exclusive or near-exclusive tool types of the Early Aurignacian (i.e., Aurignacian I) [12,15,19,43,141] but are also known from Proto-Aurignacian assemblages at Grotte du Renne, Les Cottés, Grotta di Fumane, and Labeko Koba [30,90,142,143]. They feature a lateral, direct, and scalar retouch occasionally described as 'Aurignacian retouch' and/or accompanied by a lateral notch or 'strangulation' [144]. Therefore, in our dataset, *AurignacianBlades* represent artefacts specifically

described as 'Aurignacian blades' or 'notched/strangulated Aurignacian blades', with descriptions of blades with 'Aurignacian retouch' considered too ambiguous for inclusion. Likewise, retouched blades solely described as 'strangulated' are represented by the *RetouchedBlades* category as a conservative recording of the ambiguous data.

Dufour bladelets, which feature an inverse or alternate marginal and semi-abrupt retouch on lateral edges [132], are known to be diversely represented across the different chrono-cultural variants of the Aurignacian. The two defining subtypes are often viewed as time-sensitive, with Dufour subtype Dufour bladelets being important components of Proto-Aurignacian and Early Aurignacian toolkits – albeit produced by different technological schemes [21] – and the smaller, twisted Dufour bladelets of the Roc-de-Combe subtype being characteristic of the Evolved Aurignacian. *Font-Yves* bladelets are pointed bladelets shaped by a bilateral and direct retouch of a marginal extent like Dufour bladelets [132]. They are equivalent or near-equivalent to the Krems points of Central and Eastern Europe [49,141,145] but often described as Font-Yves points by Western European researchers after the eponymous French site [146]. As Dufour bladelets and Aurignacian blades are conventional index fossils of the technocomplex, they are particularly important for assessing Aurignacian variability and the application of classical definitions to the material record of different regions, such as Iberia.

Miscellaneous retouched tools described as 'pieces' rather than blades, bladelets or flakes and without a clear typological classification are represented by the *Ret.MISC* category. Some common descriptions in the literature that are relevant to this category include, but are not limited to, "atypically retouched pieces", "naturally backed knifes" and "diverse/various" pieces. As counts for notches and denticulates are not always reported separately in publications, they here fall under the single group of *NotchesDenticulates*. The *SplinteredPieces* category refers to what is understood as *piezas astilladas* and *pièces esquillées* in the Spanish and French lexicon. Typologically, these intermediate tools were traditionally interpreted as having scars that originate from percussion, presumably from their use as wedges [144], but such scars can also derive from bipolar reduction (i.e., from their function as cores via percussion on anvil). Previous studies have drawn attention to some of their characteristic features as tools and cores [e.g., 147] but as noted above, the data have been recorded in accordance with the terminologies used in publications. We therefore make no interpretation-based transformations of the data during the recording process; in other words, whether such pieces are splintered pieces, bipolar cores, or both.

A small subset of typological attributes consists of retouched tools that are less characteristic of the Aurignacian. These include: *Microgravette* points, *ChâtelperronianPoints*, and *MP.Points*, the latter of which combines 'retouched Levallois points' and 'Mousterian points' (S1 Fig).

## Chronological and geographic data

Geographic and chronological data are used here to calculate spatial and temporal distances between assemblages, thereby allowing the spatio-temporal dimension of variability to be tested with partial Mantel tests (see 'Mantel tests' subsection below). Chronological data in the form of radiocarbon and luminescence dates have been obtained from published literature (S2 Table). Radiocarbon dates were calibrated in R (version 4.2.3) [148] with the package 'rcarbon' (version 1.5.1) [149] using the IntCal20 and Marine20 curves [150,151]. For the two assemblages with marine shells (i.e., Bajondillo level 11 and Gorham's Cave level CHm.5), reservoir offsets were obtained from the Marine20 database [150]. As the calculation of temporal distances requires a representative age for each assemblage, we used the calculated median given by the 'binMed' function in 'rcarbon' [149]. Although various methods exist to summarise assemblages with a single age; for example, by obtaining a random date from the probability distribution of the derived calibrated date [152] or averaging a group of median dates [153], median ages provide a straightforward approach.

Site coordinates necessary for calculating spatial distances were obtained from publications or official heritage databases curated by state or regional authorities. In limited cases where coordinates were not published, site locations provided on maps were cross-referenced with web-based GIS platforms and the coordinates were subsequently extracted. Site coordinates used in our analyses were formatted under the WGS 84.

## Multivariate statistical analyses

**Correspondence analysis.** The first component of our analyses entails the use of Correspondence Analysis (CA) to explore inter-assemblage variability in multivariate space. CA is an exploratory multivariate technique equipped for categorical data and therefore well suited for presence-absence datasets [154,155]. In archaeology, CA is particularly useful for exploring relationships between observations and variables, including the potential clustering of groups in multivariate space [120,156]. As such, we performed CA to explore the structure of the lithic assemblage dataset with the objective of examining the degree of patterning among chrono-cultural variants and the association between assemblages and techno-typological attributes. This was carried out in R using the 'factoextra' package (version 1.0.7) [157].

**Mantel tests.** The second component of our analyses involves the use of partial Mantel tests [158] to test whether assemblage dissimilarities are related to spatial and temporal distances. Mantel tests compare two independent distance matrices under the null hypothesis that distances among objects in one matrix are not linearly related to distances in a second matrix [159]. Partial Mantel tests [160] perform the same analysis whilst controlling for the effect of a third distance matrix. This is particularly beneficial in cases where autocorrelation may be expected between two distance matrices. However, as it has been argued that partial Mantel tests may not completely remove the effects of autocorrelation, therefore leading to Type I errors from elevated *p*-values [161], we also combined Mantel tests with a Moran's Spectral Randomisation (MSR) procedure [162,163]. This technique uses a spatially constrained randomisation procedure to remove the effect of spatial autocorrelation from the statistic of a simple Mantel test. In our application of the MSR procedure, we ran separate tests with spatial weights of five ($K=5$) and ten ($K=10$) nearest neighbours to examine the effect of increased neighbourhood sizes.

In archaeology, Mantel tests have been computed with various distance matrices to great effect, but it is clear that spatial and/or temporal distances are variables of a certain primacy, not least because of the discipline's long-standing interest to understand behavioural change across space and time [e.g., 153,154,164–167]. Likewise, the three matrices computed in our partial Mantel tests are: Jaccard distance, spatial distance and temporal distance.

Jaccard distance refers to a quantification of inter-assemblage dissimilarities using Jaccard's coefficient. Jaccard's coefficient ignores negative matches (i.e., shared absences of attributes between assemblages) whilst calculating a distance measure based on positive matches. This makes it highly appropriate for presence-absence datasets given that such absences may reflect sampling error. Values for Jaccard's coefficient range between 1 and 0, with 1 being a complete absence of shared attributes and 0 an identical presence.

The spatial distance matrix represents the geodesic distance between each assemblage based on latitude and longitude coordinates. A geodesic distance accounts for the curvature of the Earth and is therefore more suitable for measuring extensive distances between points, as in this case. As a sizeable proportion of assemblages derive from the same site, the absence of spatial distance between assemblages from the same stratigraphic sequence may influence the results of partial Mantel tests that incorporate spatial distances. Moreover, some lithic artefact types are known to be geographically clustered (e.g., Aurignacian blades, which are only known from northern Iberian sites) rather than randomly distributed. Therefore, the application of MSR to the Mantel test is necessary to account for these spatial associations [163].

Our third distance matrix, temporal distance, represents a simple Euclidean distance between the median age of each assemblage. Median ages were therefore calculated using calibrated radiocarbon dates bar two cases where only luminescence dates are available (i.e., Gato Preto and Cardina-Salto do Boi). Naturally, as temporal distances can only be calculated with dated assemblages, analyses with partial Mantel tests encompass 37 assemblages as four assemblages from the sample analysed with CA are undated: Finca de Doña Martina level 8, Lapa do Picareiro level HH, Labeko Koba level III and Aitzbitarte III level Vb-base. While not ideal, this demonstrates the imperfect nature of Palaeolithic data and the challenges attached to broad-scale quantitative analyses. Nonetheless, as the exclusion of four assemblages is minimal it does not compromise the ability to draw meaningful inferences about the spatio-temporal variability of Early Upper Palaeolithic assemblages.

Partial Mantel tests were carried out in R with Spearman's correlation coefficient – given its suitability to non-parametric data – and 9999 permutations of the data using the 'vegan' package (version 2.6.4) [168]. Jaccard's coefficient was calculated with the 'vegdist' function of the same package, whilst geodesic distances between assemblages were computed with the 'geosphere' package (version 1.5.18) [169]. The application of MSR to the Mantel test was carried out with packages 'ade4' (version 1.7.22) and 'adespatial' (version 0.3.23) [170,171] following Crabot et al. [163]. In the interest of scientific reproducibility, the data and corresponding R script used to perform the statistical analyses have been made available in the accompanying S1 File and Zenodo repository (https://doi.org/10.5281/zenodo.18462036).

## Results

### Correspondence analysis

The first four dimensions of the CA together account for 43% of the variation in the dataset of lithic assemblages, indicating a degree of complexity in the relationships between variables that is not easily summarised by dimension reduction (Fig 3). On the basis that some characteristically non-Aurignacian attributes (i.e., *LevalloisTech, ChâtelperronianPoints, MP.Points* and *Microgravettes*) may be introducing unwanted noise, this value only increases to 46% when these attributes are removed from the analysis (S2 File). In part, the low score of the first four dimensions may be attributable to the large number of columns processed by the CA. Furthermore, as the first two dimensions only account for 25% and 27% of the variation under each respective analysis, there is an additional indication that the dataset is heterogeneously

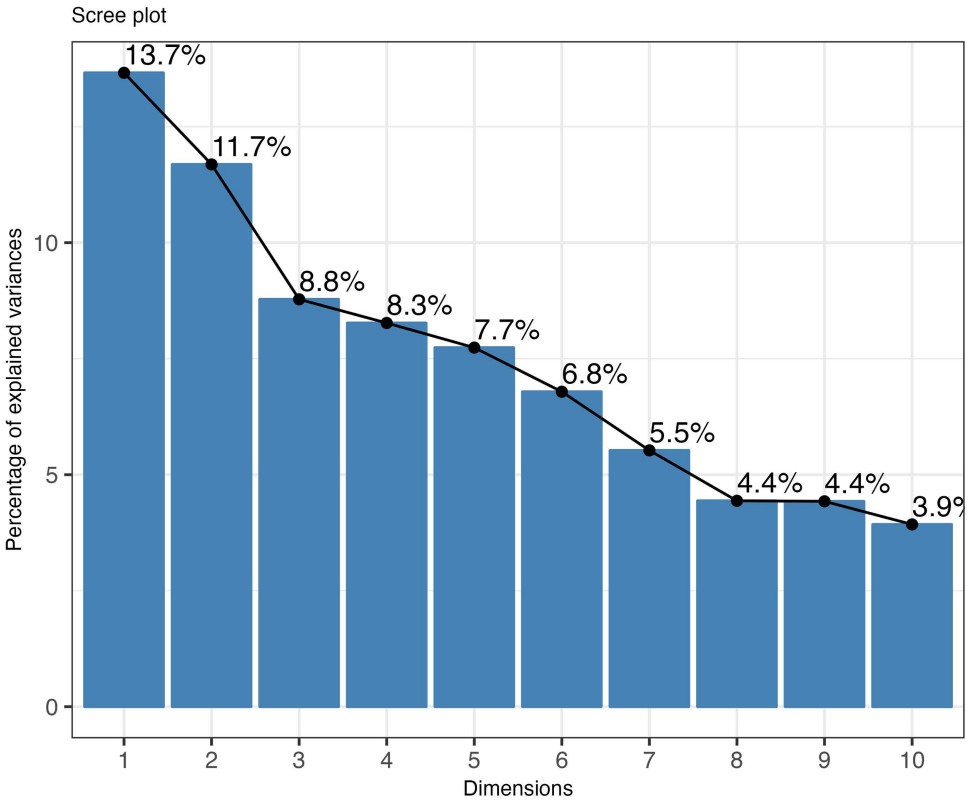

**Fig 3. Scree plot showing the contribution of each dimension to the explained variance.**

structured (Fig 3 and S2 File). This is not at odds with the diverse representation of techno-typological attributes across assemblages, given that in some cases absences outweigh presences (Fig 2).

Nevertheless, a row plot of the assemblages shows that the first dimension – the most significant one extracted by CA – broadly separates assemblages from northern areas (positive values) and southern areas (negative values) along its axis with a few exceptions: Gorham's Cave level CHm.5, Bajondillo level 11 and Gato Preto level C, with positive coordinates, and Cova Gran level 497D, Labeko Koba level VI, Cova Foradada level IIIc and Cardina-Salto do Boi level G5-A10 with negative coordinates (Fig 4A). In this regard, northern assemblages (meaning those with latitudes greater than 40˚N) show a stronger degree of overlapping in multivariate space than southern assemblages, indicating that associations between them are stronger. Conversely, differences among southern assemblages are generally more pronounced, which is a pattern made clear by the wider distribution of assemblages in multivariate space and strongly influenced by the placement of Cova Malladetes assemblages in the upper left corner of the plot. Considering the underlying geographic context, assemblages from the same site do show some relatedness although the strength of these associations varies significantly; for example, the intensity of associations between assemblages from Cova Malladetes and La Viña are markedly different.

In terms of the association of assemblages and techno-typological attributes, a biplot of rows and columns (Fig 5A) demonstrates a rather diverse structure characterised by notable examples of overlap, which is perhaps again attributable to the quantity of variables and the fact that the first two dimensions account for 25% of the variance. Nevertheless, one clear observation is that certain attributes seem to be less discriminating among assemblages, as per their central position in the plot (e.g., *BladeBlanks, NotchesDenticulates*, *Endscrapers*, *DufourDufour* and to a lesser extent, *Carinated.EndscraperCores*, *Burins* and *CoreWorking* elements). On the other hand, some attributes appear to be rarer cases that are more distinct to certain assemblages, as indicated by their position at the extremities (e.g., *ChâtelperronianPoints*, *MP.Points*, *LevalloisTech*, *Ret.Flakes*, *FlakeCores*, *DufourRoc.de.Combe*). Curiously, *DufourRoc.de.Combe* bladelets have elevated positive coordinates across both dimensions despite being known as index fossils of the Evolved Aurignacian, which is the most numerous chrono-cultural variant in the studied dataset. This differs to *DufourDufour* bladelets, whose near-central position suggests they are a less discriminating variable. As shown in the binary heatmap, *DufourRoc.de.Combe* bladelets are a poorly represented artefact type, even among Evolved Aurignacian or Late/Evolved Aurignacian assemblages (Fig 2).

Considering the above, the biplot also reveals far more discrete retouched tool types to the right of the central axis, where northern assemblages are seen to cluster. On the contrary, assemblages to the left of the axis (which are predominantly southern assemblages) are more strongly associated with generic artefact classes such as *FlakeBlanks*, *BladeletBlanks*, *Ret.Bladelets* and *Ret.MISC*. By extension, it is not surprising that *Ret.Blades* and *AurignacianBlades* are nested among northern assemblages close to the positive axis of the first dimension, given that such tool types are synonymous with the Early Aurignacian – a chrono-cultural variant disproportionately represented in northern Iberia. However, in this regard, it is curious that Gato Preto level C (Evolved Aurignacian) appears closely related to these variables and other northern assemblages represented by earlier phases of the Aurignacian, e.g., El Castillo level 18C, L'Arbreda level H, Labeko Koba level VII, Cueva Morín level 8 and La Viña level XII.

While there is an indication that certain techno-typological attributes are more associated with some assemblages over others (Fig 5A), they contribute to the variation expressed by the first two dimensions in differing amounts (Fig 5B). As shown in Figs 6A and 6B, the variation explained by the first two dimensions is driven by an array of different variables with relatively small contributions, except for the loading of *DufourRoc.de.Combe* bladelets on the second dimension. In other words, there is no group of variables which account for a large proportion of the variation explained by each of the first two dimensions. In the case of the first dimension, the contributions of these variables are not much greater than the value expected (red dotted line) if the collective contributions were uniform [157]. When the contributions to the first two dimensions are considered together, *DufourRoc.de.Combe* bladelets are seen to account for 17.5% of the captured variability, with the rest of variables showing smaller contributions under 7.5%, albeit still above the expected values (Fig 6C).

                                                                   

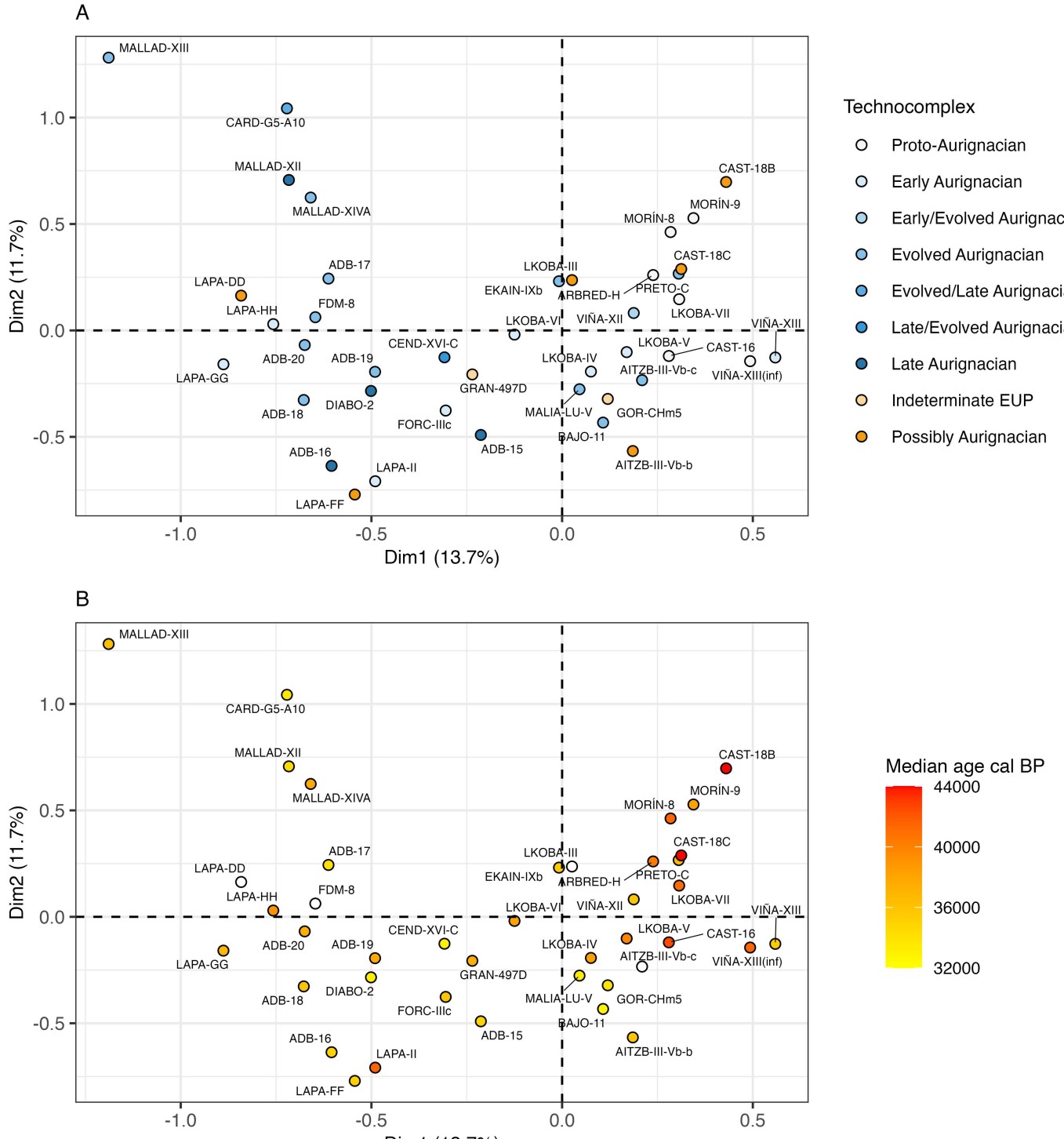

**Fig 4. Correspondence Analysis row plots of the first two dimensions.** A) Row plot of assemblages coloured by technocomplex and B) row plot of assemblages coloured by median calibrated ages BP (the four undated assemblages are coloured white).

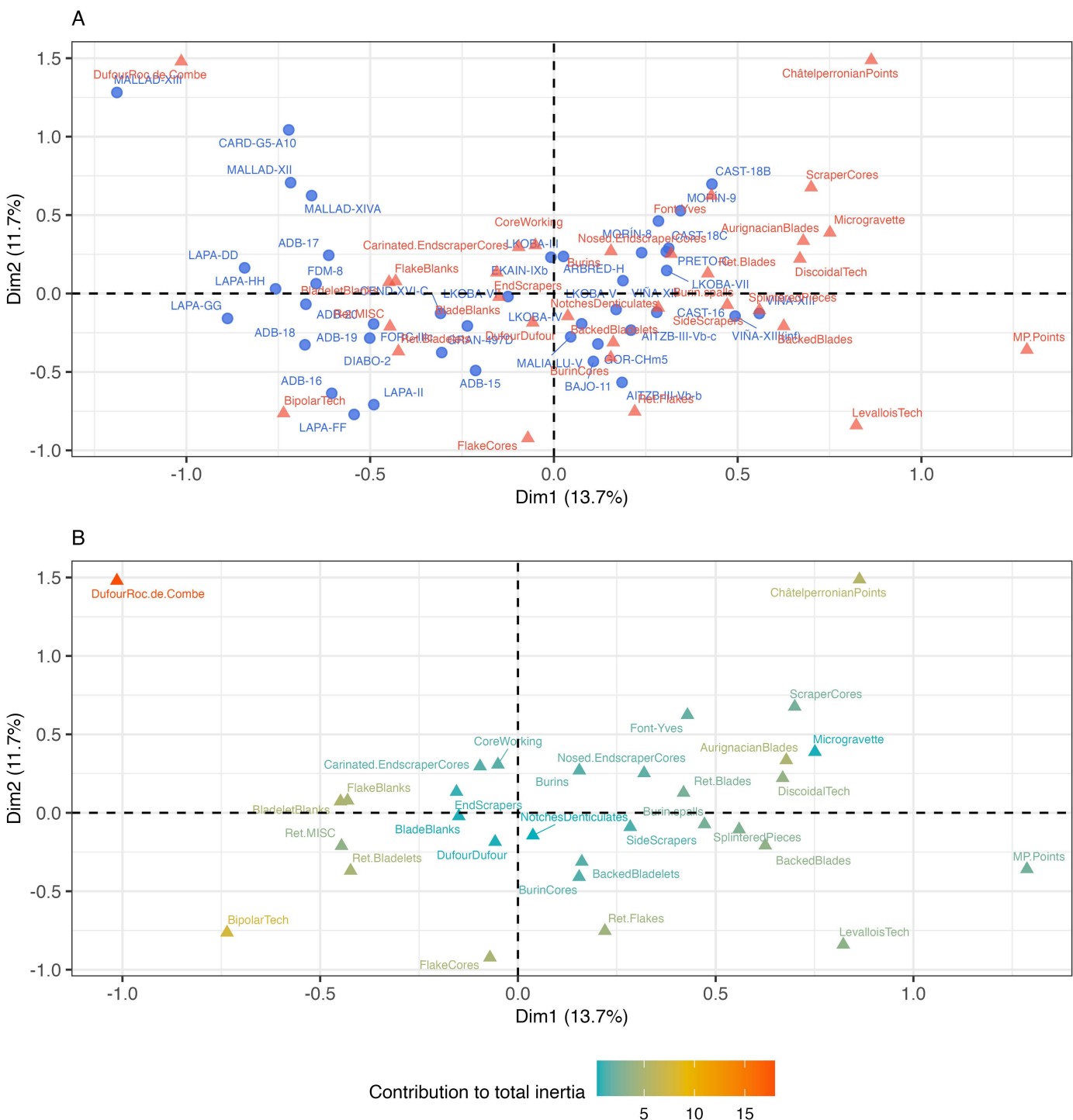

**Fig 5. Correspondence Analysis biplot and column plots.** A) Combined biplot of rows (assemblages) and columns (techno-typological attributes) and B) column plot showing the contribution of variables (techno-typological attributes) to the variance explained by the two dimensions. The colour ramp indicates the contribution as a percentage.

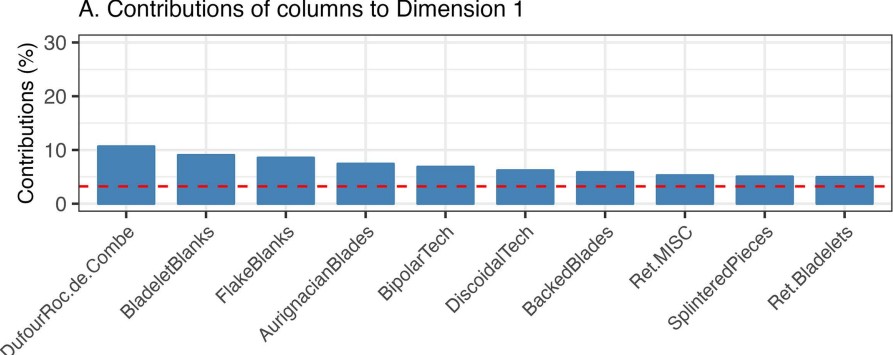

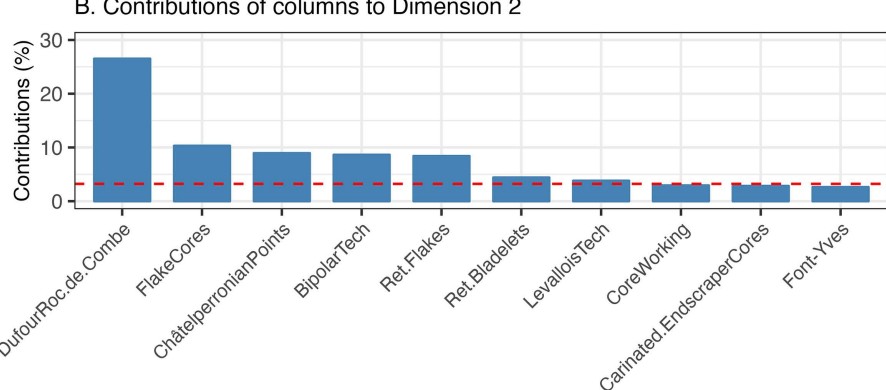

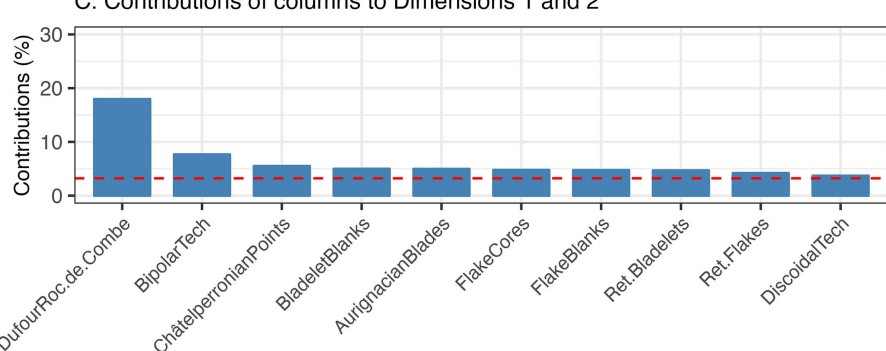

**Fig 6. Bar graph showing the contributions of column variables (techno-typological attributes) to the first two dimensions.** A) Contributions to the first dimension, B) contributions to the second dimension and C) Combined contributions to the first and second dimensions.

Finally, although four assemblages are undated, the remaining sample does not show any strong temporal patterning of the Aurignacian sub-variants. For instance, while some assemblages of similar ages show a degree of close association, there are numerous examples of overlap by assemblages with diverse relative ages, e.g., Cova Foradada level IIIc with La Boja level 15, or Abrigo de la Malia level LU-V with Labeko Koba levels V and VI. This is more visible when the assemblages are highlighted according to their median calibrated ages BP (Fig 4B). The non-uniform character of this chrono-cultural distribution is also seen in the fact that Proto-Aurignacian assemblages (n = 6) are closely related to

Early Aurignacian assemblages from similar northern latitudes, but not to the reported Early Aurignacian assemblages found further south (Lapa do Picareiro and Cova Foradada), which plot to the left of the central axis. Concerning the Evolved Aurignacian, assemblages are not grouped by a single cluster but widely distributed across multivariate space. As the most abundant variant (n = 12), this pattern may result from the greater diversity and spatial distribution of its lithic assemblages.

Undiagnostic assemblages (i.e., 'Possibly Aurignacian' and 'Indeterminate EUP') do not show any strong clustering based on the sample analysed (n = 8). However, some show an expected association to nearby sites or assemblages from the same sequence (Lapa do Picareiro levels DD and FF, Labeko Koba level III, and El Castillo level 18C). On the other hand, others do not meet this expectation but are associated with assemblages from diverse regions (Cova Gran level 497D, Gorham's Cave level CHm.5 and Aitzbitarte III level Vb-base). This supports the idea that their lithic assemblages are more distinct and perhaps dissimilar to nearby sites.

## Partial Mantel tests of distance matrices

As CA is an exploratory multivariate technique that can display general patterns and associations, Mantel tests provide an opportunity to test the explanatory relationships (if any) behind these patterns. Our partial Mantel tests reveal a moderate, positive correlation (r = 0.35, $p$-value = < 0.001) between Jaccard distance and spatial distance, demonstrating that lithic assemblage dissimilarities increase with spatial distance (Table 2). However, as outlined earlier, the significance of this relationship may be influenced by some spatial autocorrelation, the effect of which partial Mantel tests may not fully eliminate [161]. To a certain degree, some spatial autocorrelation is to be expected because assemblages located close together – including those from the same site – are often more similar. However, as this expectation violates the assumption of independence between distance matrices in a Mantel test, a more accurate reading is necessitated by computing the Mantel test with MSR (see 'Materials and methods' section) [163]. This provides a corrected level of significance ($p$-value = 0.0052 at K = 5 and $p$-value = 0.0172 at K = 10) that corroborates the correlation between Jaccard and spatial distance after the effect of spatial autocorrelation is removed (Table 2).

Partial Mantel tests between Jaccard distance and temporal distance provide an entirely different result, showing no correlation between the two matrices (r = −0.02059, $p$-value = 0.566) (Table 2). This result is also replicated when more stringent criteria are used in the calculation of assemblage ages, such as the selection of radiocarbon samples processed with more rigorous pretreatment methods (S2 File). The absence of this linear relationship therefore demonstrates that assemblage dissimilarities do not increase with time. This could indicate that techno-typological changes are subtle or even negligible across the relevant timeframe (c. 43–32 ka cal BP), at least with respect to the more coarse-grained resolution of variability provided by the binary lithic dataset. On the other hand, the absence of a linear relationship may point towards an underlying, nonlinear pattern of inter-assemblage dissimilarities that are not detected by partial Mantel tests. In any case, inter-assemblage dissimilarities do not appear to be temporally expanding phenomena, suggesting that alternative patterns of temporal change (if any) may be relevant.

**Table 2. Results of partial Mantel tests between the three distance matrices.**

| Distances matrices | Statistic | $p$-value | MSR $p$-value | |
|---|---|---|---|---|
| Jaccard distance and spatial distance, controlling temporal distance | 0.35 | **<0.001** | **0.0052[a]** | **0.0172[b]** |
| Jaccard distance and temporal distance, controlling spatial distance | −0.02059 | 0.566 | – | – |

The MSR p-value is a corrected level of significance following the application of the Moran's Spectral Randomisation procedure to a simple Mantel test between Jaccard and spatial distance.

a Following spatial weights based on K = 5

b Following spatial weights based on K = 10

## Discussion

### The spatial structure of inter-assemblage variability

The idea that the Aurignacian is a spatially and temporally variable phenomenon has been mentioned by those seeking to emphasise the diverse and inconsistent representation of cultural and technological behaviours during the Early Upper Palaeolithic [52,61,145]. This emphasis on mosaicism diverges from a more culture-historical understanding of the Aurignacian as a technocomplex with distinct, identifiable traits, and with a distribution and chronology that documents a westward expansion of modern humans across Europe [44]. However, quantitative approaches to lithic assemblages have rarely informed debates on the 'compositional integrity' of the Aurignacian [172], and even less so on the significance of undiagnostic assemblages within this inferred mosaic.

Regarding the spatial variation of the material record, our results prove that inter-assemblage variability has a significant spatial component in the Iberian Peninsula. This is indicated by the regional patterning of assemblages in a CA row plot, including the broad separation of north and south assemblages across the first dimension, and further corroborated by Mantel tests of the relationship between assemblage dissimilarities and spatial distances, which provide convincing evidence for the spatial structure of variability. Taken together, both analyses suggest that the geographical location of Iberian assemblages is a strong determinant of techno-typological variability and that, as a corollary, inter-assemblage proximity is a predictor of techno-typological similarities (*sensu* Tobler's [173] Law of Geography).

The relationship between inter-site distances (both spatial and temporal) and artefact variability is a relationship of fundamental interest in archaeology, not least because it can hint at the existence and effect of social networks, population structures and changing environmental conditions. Where Mantel tests have been used to explore inter-assemblage variability, a spatial correlation has sometimes been attributed to an 'isolation by distance' effect [174] in which artefact variation reflects decreasing effects of interaction and exchange [153,175]. In Iberia, low levels of interaction and exchange may be one explanation for the spatial component of inter-assemblage dissimilarities during the Early Upper Palaeolithic. However, this would contradict the wider European evidence of communication and exchange shown by the clustering of Aurignacian personal ornaments [167,176]. In this sense, it is reasonable to suggest that lithic and osseous similarities among regionally bound sites, such as those in Cantabria [177], do indeed reflect flows of information. Likewise, in Catalonia, the potential long-distance movement of flint in the Proto-Aurignacian of L'Arbreda (level H) from a French source located 100 km away [116] could not just be the result of mobility, but also exchange within networks. However, at present, studies of forager networks during the Iberian Aurignacian remain relatively peripheral [but see 178–180] in part because the number of Aurignacian sites is significantly lower than the subsequent Gravettian and Solutrean [99,181]. In any case, given that there is a positive correlation between spatial distance and variability across the entire peninsula, it is possible that existent networks were geographically and demographically confined and, where active, did not always result in the stable transmission or use of technocomplex-defining artefacts. A patchy representation of Aurignacian index fossils is particularly evident in assemblages from southern and eastern sites like Bajondillo, Gorham's Cave, La Boja, Cova de les Malladetes and Cova de les Cendres (Fig 2).

Alternative – and perhaps more falsifiable – hypotheses for the spatial dimension of inter-assemblage variability may therefore rest in the well-established link between landscape variation and forager behaviour. In particular, with respect to how distributed resources (i.e., lithic and faunal resources) influence land-use strategies and toolkit construction [182,183]. The non-uniform distribution of raw materials (of differing quality and abundance) is known to have a significant effect on assemblage composition and may therefore explain some of the techno-typological variability seen across Iberia [184–186]. In Cova Gran level 497D, the absence of Dufour bladelets and carinated/nosed end-scraper cores, coupled with a retouched tool component dominated by sidescrapers, notches and denticulates, has been partly attributed to the local fissure- and impurity-laden chalcedony that comprises 80% of the assemblage of >6000 artefacts [117,187]. Lithic refits of level 497D have shown that blade and bladelet production were interlaced with the production of flakes, the latter

of which overwhelmingly dominate the assemblage [117,188]. A similar case can be seen in Cardina-Salto do Boi level G5-A10, where the Evolved Aurignacian assemblage of local and regional quartz (94% of the lithic assemblage) is dominated by flakes, with only three retouched tools [71]. Here, blade and bladelet blanks are scarcely represented (six pieces altogether), although Font-Yves and Dufour, subtype Roc-de-Combe bladelets comprise two of the retouched tools, both being produced on brown jasper from an unknown source [179]. At Cova Eiros (Galicia), the quartz-based industries of the archaeological sequence include an Early Upper Palaeolithic assemblage dated to 36–35 ka cal BP but with no cultural diagnostics [189]. Despite this, the assemblage has been tentatively attributed to the Evolved Aurignacian based on its laminar component and age.

Recent experimental work on the mechanical performance of different raw materials has corroborated that flint is an optimal material for tool effectiveness and durability [190]. The influence of raw material quality, abundance and proximity to occupation sites may therefore be an important element in the spatial structure of inter-assemblage variability [e.g., 122], but it is not an all-determining variable. At Abrigo de la Malia, although diagnostic artefacts are absent from the presumed Evolved Aurignacian assemblage of level LU-V, a carinated burin core in hyaline quartz demonstrates that bladelet production was not impeded by the selection of other – potentially less preferred – materials (all the five retouched tools are produced on flint) [73]. Furthermore, examples of blade/bladelet production on coarse-grained rocks are commonly known from the Middle Stone Age of South Africa, proving that diverse raw materials can still be adequate for laminar products [e.g., 191–193]. More provenance data is therefore needed to establish how raw material distributions may shape the spatial dimension of inter-assemblage variability across Iberia. In this regard, the related concepts of curated and expedient technology [183,194] are important lines of inquiry that should be pursued with large comparative datasets. Low-cost and expedient technological strategies could be responsible for the less diagnostic assemblages known from different regions of Iberia, resulting in the paucity or absence of standardised and technocomplex-defining implements [195].

Although we are outlining behavioural factors of spatial variability, we also wish to highlight that techno-typological variability is a multicausal phenomenon in which taphonomic processes play a significant role – a case made clear by vertical refits from the Cardina-Salto do Boi sequence [71]. Nevertheless, behavioural explanations may have a certain currency given that in several multi-layered sites, raw material selection and use is largely consistent while the composition of artefact assemblages is not (see also S2 Fig for a visualisation of inter-assemblage dissimilarities). Under such conditions, variability could result from changing activities and mobility strategies as previously demonstrated in the Aurignacian of southwest France [196]. In Labeko Koba (Basque Country), imported flint from sources 30–70 km away comprises 99% of lithic materials in levels III–VII, but the size and techno-typological composition of assemblages varies throughout, including the representation of conventional Aurignacian index fossils (Fig 2) [90,178]. For instance, the comparatively smaller and less rich assemblages of levels III and V contain no Dufour bladelets, Aurignacian blades or carinated and nosed end-scraper cores; but even in the larger Early Aurignacian assemblage of level IV (6323 lithics, of which 268 are retouched), Dufour, subtype Dufour bladelets are limited to one example. An analogous example from south-east Iberia is provided by La Boja's archaeological sequence, where the predominant use of flint in the six Aurignacian levels accompanies shifting assemblage compositions and occurrences of typological index fossils (Fig 2) [67]. To illustrate with one example, the Evolved Aurignacian assemblage of level OH19 contains no Dufour, subtype Roc-de-Combe bladelets, even though 38% of bladelet blanks were extracted from carinated or nosed end-scraper cores [67]. Clearly, therefore, occupation duration, artefact discard/loss and the off-site transport of implements (operating under site-scheduled activities and wider mobility patterns) are processes that likely drive the compositional variability of the Iberian record, even when other variables are held constant.

Within this framework, short-term occupations are a particularly interesting phenomenon that could underpin the smaller, less diverse (and in some cases, less diagnostic) assemblages known from Iberia. If shorter lengths of stays are expected to result in less lithic material – although not necessarily from a reduction of on-site tasks [see 197] – they can be logically attributed to a poorer representation of techno-typological traits. In line with the well-known relationship

between assemblage size and richness [198,199], smaller quantities of material can therefore be equated to less heterogenous assemblages and a corresponding lower probability that culturally diagnostic elements are found. A natural consequence of this is that from a cultural taxonomic perspective, small assemblages may not always display the defining characteristics of a technocomplex. However, from a behavioural perspective, their possible relationship to short-term occupations can provide important context to the variability of known assemblages, in turn generating implications for the detection and characterisation of the Aurignacian in Iberia.

All things being considered, there are numerous variables and phenomena that shape the character and composition of lithic assemblages, many of which can be related to the distribution of resources and their corresponding exploitation. Our studied sample provides quantitative evidence for spatial distance being a significant correlate of inter-assemblage variability in the Early Upper Palaeolithic of Iberia. Therefore, as dissimilarities increase with spatial distance, assemblage variability can be considered as a logical expectation across different areas and regions. In the wider European context, this empirical finding contradicts the application of the Aquitaine model to Iberia and aligns with similar conclusions from other regional studies [31,36,58].

## Temporal variability: Absent or amorphous?

Whereas space is demonstrated to be an important variable of inter-assemblage variability, our multivariate methods provide no evidence of a temporal effect; the distribution of assemblages in the CA row plot does not show any clear clustering of Aurignacian chrono-cultural variants, while Mantel tests confirm that temporal distance is not linearly related to assemblage dissimilarities. In other words, dissimilarities between analysed assemblages do not increase with time.

This may seem at odds with the idea that techno-typological changes between Early Upper Palaeolithic assemblages are diachronically evident. After all, temporal change is the backbone of proposed differences between Aurignacian sub-variants as per the classic sequences of La Ferrassie and Abri Castanet [12,13,15,200]. However, evidence of diachronic change in context-specific sequences does not mean that such temporal patterns will be replicated elsewhere, especially in regions where long and continuous sequences of occupation are rare (i.e., spanning the entire Aurignacian). To date, there is still no Iberian sequence that documents all four conventional variants of the southwestern French Aurignacian: among multi-occupation sites of the north, the Evolved Aurignacian is always succeeded by the Gravettian (e.g., La Viña, Aitzbitarte III, L'Arbreda), while in the south, unambiguous evidence of the Proto- and Early Aurignacian is absent while a Late Aurignacian phase is implied (e.g., La Boja, Cova de les Malladetes and Cova Cendres) [67,69,70,91,103,124,201]. The idiosyncrasies of the Iberian record therefore serve to remind us that although type-sites and reference sequences can be models of diachronic change, their predictive value needs to be evaluated against the influence of context-specific phenomena and spatially mediated variables.

The absence of a temporal pattern is not, however, a simple demonstration that lithic assemblages of the Iberian Early Upper Palaeolithic are temporally consistent. For example, differences between temporally separated assemblages of the Proto- and Evolved Aurignacian can often appear quite clear, most notably in terms of blade/bladelet production, retouched bladelet morphologies and burin diversity [21,27,29,54,202]. However, our traits-based analysis indicates that, overall, temporal variation of techno-typological attributes is too unstructured to be detected by Mantel tests – it is important to reiterate here that Mantel tests are tests of linear relationships between two distance matrices, meaning that punctual or nonlinear change may be present when linear change is not. Moreover, from a methodological point of view, this negative result may be influenced by the more coarse-grained resolution of variability provided by the binary format of techno-typological data. An evaluation of the frequencies and proportions of techno-typological features could demonstrate a linear pattern of temporal variability connected to structured diachronic change. Indeed, the proportions of different core and bladelet types are said to be important markers of Aurignacian sub-variants [145]. Therefore, broad comparative analyses of continuous variables must be the pursuit of future research if we are to understand whether temporally increasing differences between Iberian assemblages (and especially Aurignacian sub-variants) are statistically supported.

Notwithstanding this, and beyond the above-mentioned effect of spatially dependent variables and assemblage size variations, there is good reason to relate the negative detection of temporal patterning to the recurrent and heterogeneous distribution of traits *across* and *within* temporal classes of assemblages, the latter being an indication of synchronic variability. Regarding the chrono-cultural variants of the Aurignacian, a degree of recurrency has been argued for the Proto- and Early Aurignacian of northern Iberia based on the co-occurrence of diagnostic artefacts like Aurignacian blades and carinated end-scraper cores [33], while heterogeneity within temporal classes has been claimed for the Evolved Aurignacian of Cantabria on account of its typological diversity [203]. In general, the diversity of assemblages immediately following the Early Aurignacian has often made it difficult to argue for a single, cohesive chrono-cultural phase [e.g., 18,60] and in northern Aquitaine, inter-site variability is reported to peak with the *Aurignacien Récent* (i.e., Evolved Aurignacian) [202]. Within our studied sample, dual classifications like 'Early/Evolved Aurignacian', 'Evolved/Late Aurignacian' and 'Late/Evolved Aurignacian' convey that the Evolved Aurignacian can be a materially ambiguous (rather than easily recognised) variant in Iberia.

Similarly, our analysed dataset lends support to an unsystematic representation of diagnostic traits across and within Aurignacian variants (Fig 2). For instance, the co-occurrence of Dufour, subtype Roc-de-Combe bladelets (an index fossil of the Evolved Aurignacian) among Proto- and Late Aurignacian assemblages is evident in Cueva Morín and Cova Malladetes respectively [70,102], whereas Dufour, subtype Dufour bladelets are found among various Evolved and Late Aurignacian assemblages (e.g., Aitzbitarte III, Pego do Diabo, La Boja, Finca Doña Martina and Cova de les Cendres) (Fig 2) [67,69,89,103]. Moreover, intra-variant variability vis-à-vis the occurrence of diagnostic artefacts is a notable feature of the Evolved Aurignacian due to the fact Dufour, subtype Roc-de-Combe bladelets are present in only 33% of assemblages exclusively classified as 'Evolved Aurignacian'.

Culturally undiagnostic assemblages are also important components of synchronic variability by virtue of their non-conforming compositions. Two clear examples are Cova Gran level 497D, an assemblage contemporaneous with the Evolved Aurignacian but lacking any variant-specific artefacts [93] and Labeko Koba level III, an undiagnostic and undated assemblage directly overlying an Early Aurignacian horizon [204]. Similarly, Lapa do Picareiro levels DD and FF and Gorham's Cave level CHm.5 are coeval with the Evolved and Late Aurignacian of southern Iberia yet constitute artefact-scant assemblages with no diagnostic traits [72,94]. However, the extent to which these smaller assemblages stem from behavioural, taphonomic or sampling processes is not fully clear. For instance, Barton and Jennings [94] suggested that Gorham's Cave level CHm.5 could equate to level D of Waechter's excavation, a deposit that was divided into two sub-units containing 290 and 278 lithic artefacts [205,206]. However, this potential connection is extremely difficult to demonstrate given that – issues of stratigraphic correlation aside – the locations of only 31 artefacts from level D are currently known [94]. Nevertheless, based on the fact that Cova Gran level 497D contains more than 6000 lithic artefacts excavated over 55 m² [117], there is reason to suspect that sampling bias alone cannot fully explain the characteristics of both undiagnostic and diagnostically poor assemblages and that, as a result, such assemblages may constitute meaningful examples of behavioural variation in the regional Aurignacian timeframe.

Regarding the technological evidence – and similar to what Tafelmaier observed [33] – carinated and nosed end-scraper cores are not consistent features of Early or Evolved Aurignacian assemblages and can additionally be found in Proto- and Late Aurignacian assemblages, e.g., El Castillo level 16, L'Arbreda level H, Labeko Koba level VII, Cueva Morín level 8, Cueva Morín level 9, La Viña level XIII(inf.) and Malladetes level XII. On a quantitative basis, elevated numbers of carinated cores have not only been detected in the Proto-Aurignacian of L'Arbreda [116] but also Grotta di Castelcivita in southern Italy [32]. These observations and findings deviate from a strict interpretation of the classic chrono-cultural model of lithic change, although the Late Aurignacian continues to be a poorly defined phase in Iberia and beyond [27,67].

In addition, we consider it significant that examples of bipolar, discoidal and generic flaking methods can be found across numerous and chronologically diverse assemblages in Iberia (Fig 2). Moreover, it should be noted that the reported

occurrence of bipolar knapping may be an underestimation given that many northern assemblages contain numerous splintered pieces but no bipolar cores. As highlighted in previous works, the overlapping characteristics of these artefacts can obscure the fact that they were used to obtain blanks via percussion on anvil, rather than as mere intermediary tools (i.e., splintered pieces) [147,207]. Although not traditionally seen as being synonymous with the Aurignacian, bipolar knapping is also known to feature in many Aurignacian assemblages from Italy; conversely, it appears far less frequently among French assemblages [32,208–211].

Discoidal methods similarly attest to the importance of flaking schemes within the Iberian Aurignacian. Although mainly present in sites from the Cantabrian coast, they are also known from Gato Preto level C and Cova Cendres level XVIC, assemblages ascribed to the Evolved and Late/Evolved Aurignacian respectively [69,89]. The occurrence of discoidal knapping in these later Aurignacian assemblages additionally shows that this technique cannot simply be regarded as the result of stratigraphic intrusions, as neither level directly overlies a Middle Palaeolithic one. In La Viña, discoidal methods persist in the sequence until the Early/Evolved Aurignacian of level XII whilst in Cueva Morín, they remain well represented in the most recent Proto-Aurignacian level [91,212]. As others have pointed out [213], the importance of flakes in the Early Upper Palaeolithic has been frequently overlooked even though flakes and flake cores can be sizeable components of Aurignacian assemblages [141,214,215]. In fact, discoidal and generic flaking methods show equally low contributions to the first two dimensions as attributes that are more representative of the Aurignacian, such as unretouched and retouched bladelets, meaning they are traits which may not strongly differentiate assemblages (Fig 6C). To this it is also worth adding that among our studied sample, there are several examples of retouched tool inventories containing high proportions of flakes (e.g., Cueva Morín levels 8 and 9, Cova Gran level 497D, Gato Preto level C, Bajondillo level 11) [89,102,119,188].

As is always the case, sampling approaches can be reasonable explanations for the composition of archaeological assemblages, as every excavation constitutes a spatial sample that may not expose the full range of activities and artefacts pertinent to a site, including culturally diagnostic pieces. In addition, it is also expected that taphonomic and other post-depositional processes play a part in the less conventional characteristics of artefact assemblages, such as those from the basal Aurignacian layers of Cueva Morín, La Viña and El Castillo, where Châtelperronian and Middle Palaeolithic points have been detected, albeit in small numbers [91,102,113]. These same processes often underlie the anomalous and unexpected ages of assemblages like those from Cueva Morín, where the chronology of the site has been described as the 'Achilles heel of the deposit' [92: 74]. However, keeping in mind that all assemblages are palimpsests and that, additionally, the known corpus of archaeological sites provides an incomplete picture of past behaviour, the variability of toolkits across and within temporal classes of assemblages may still reflect functional solutions to changing (or even unchanging) needs – solutions which did not always induce the production of archetypal implements. A prime example of this can be seen in the predominance of bipolar cores in the assemblages of the Evolved and Late Aurignacian at La Boja (levels OH16 and OH17) wherein additionally, one of the two Dufour bladelets from level OH16 was produced on a bipolar blank, despite bladelet cores and their associated blanks being well represented (conversely, carinated and nosed end-scraper cores are absent) [67]. Within such a functional framework, it is reasonable to consider that the absence of Dufour bladelets *sensu lato* in other Iberian assemblages may indicate a preference for alternative (but no less effective) microliths as hafted implements, such as backed bladelets, retouched bladelets or even small retouched flakes (e.g., Aitzbitarte III level Vb-base, Ekain level IX-b, Labeko Koba level IV, Cova Gran level 497D, La Malia level LU-V, Bajondillo level 11 and Gorham's Cave level Chm.5). The question therefore remains whether the absence or minimal proportion of diagnostic artefacts reflects a diversely manifested (but nonetheless established) Aurignacian or, on the other hand, low-fidelity forms that were inconsistently produced. This latter perspective would imply that the Aurignacian was not a universally shared or strongly expressed tradition in the Iberian Early Upper Palaeolithic.

All in all, referencing our results to the data leads us to propose that the co-occurrence of "diagnostic" artefacts and diverse representation of techno-typological traits underpins the absence of a linear relationship between time and

inter-assemblage differences. This aspect of internal variability is not unique to Iberia and has similarly been recognised in other regions, where the characteristics of Aurignacian sub-variants are said to be subtle rather than striking [34,216]. Under this premise, the use of index fossils to situate Iberian assemblages within the classic Aurignacian chrono-cultural scheme is a problematic approach given that such index fossils: (1) co-occur across Aurignacian sub-variants and (2) are not consistently present even when chrono-stratigraphic evidence predicts them to be (as per its diachronic expectations). In any case, too much weight is often placed on the cultural diagnosis of assemblages – and for that reason, chronological and typological evidence – at the expense of exploring the drivers of assemblage variability as evidence of relevant behavioural variation [[see 37]. As suggested here, hypotheses of functional and spatially mediated behaviour may be particularly relevant, not least because we find evidence of spatially structured variability among Iberian assemblages. At the same time, however, additional data and analyses are required to assess the structure of temporal variability during the regional Aurignacian timeframe of 43–32 ka cal BP.

## Conclusion

The emergence of widespread technological and cultural behaviours in the Early Upper Palaeolithic is commonly understood as a manifestation of the Aurignacian *sensu lato*, a technocomplex that can be traced through assemblages distributed across the entire European continent. However, although corresponding assemblages often share core characteristics, these characteristics are not universally documented; instead, artefact assemblages appear synchronically and diachronically variable in a manner that reveals an inconsistent representation of techno-typological traits [52,61]. In this broad context, the Iberian Peninsula represents one regional example defined by a rather idiosyncratic record of human settlement in which undiagnostic assemblages are found alongside diverse Aurignacian ones, the former of which are seldom factored into questions of Aurignacian variability. Moreover, while numerous aspects of the Iberian record have been debated over the past decades [80,85,99,181,217–220], broad, quantitative-based comparisons of lithic assemblages have been largely absent. Our study has therefore aimed to probe this inter-assemblage variability by analysing a newly compiled dataset of techno-typological attributes from as many Iberian assemblages as possible.

With the use of two separate multivariate techniques, we have presented a combined method of exploring associations between lithic assemblages and, furthermore, establishing whether inter-assemblage dissimilarities are related to spatial and temporal distances. Explorations of the data with CA have enabled the visualisation of relationships between lithic assemblages and techno-typological attributes, revealing a complex structure of the data that is not easily summarised. On the other hand, the distribution of lithic assemblages in a row plot reflects a spatial, rather than temporal, pattern in which northern and southern assemblages are broadly separated. The indication of spatially structured variability is corroborated by the results of partial Mantel tests which provide evidence for a linear relationship between spatial distance and inter-assemblage dissimilarities. However, this relationship does not hold for temporal distance, revealing that temporal change is not linearly detected at the traits-based resolution of our analysis.

Although the synchronic and diachronic variability of Aurignacian assemblages has been frequently mentioned [20,202,203], it has seldom been tested across extensive scales. In answer to this, our study provides confirmation that variability in the Iberian Aurignacian has a significant spatial structure, in turn suggesting that spatially mediated variables (e.g., distributed resources and the varied exploitation thereof) are important underlying factors of assemblage compositions. Conversely, temporal variability is unsupported by our analyses and may be a consequence of the heterogeneous distribution of techno-typological traits across and within temporal classes of assemblages. These aspects of variability show that the regional Early Upper Palaeolithic (inclusive of undiagnostic assemblages) constitutes a diverse, and not always uniformly structured, expression of hunter-gatherer groups. Consistent with the idea of mosaicism [61], this further implies that the Aurignacian may not be a strongly manifested tradition across Iberia. While groups were certainly connected by common ideas, the data suggests that these ideas were fluid enough to allow for behaviours

that selected for alternative approaches and responses; that is to say, approaches that did not always entail the use of technocomplex-defining techniques or implements.

Lastly, on the basis that inter-assemblage dissimilarities are found to increase with spatial distance, our findings contradict the cross-regional application of context-specific models like the Aquitaine scheme. Through this lens, the Iberian Peninsula constitutes a materially diverse record of human occupation and behaviour that should be understood on its own distinctive terms. Further work is therefore needed to establish the reasons for this behavioural variation (i.e., the drivers of assemblage variability) and the validity of taxonomic and typological systematics for understanding it.

## Supporting information

**S1 Table. Lithic assemblage dataset.** Processed binary data of techno-typological attributes.
(ZIP)

**S2 Table. Chronology dataset of processed and unprocessed dates.** Processed radiocarbon and luminescence dates are accompanied by calibrated and median ages. These reported ages were used to calculate the median assemblage ages for partial Mantel tests.
(ZIP)

**S1 Fig. Sankey diagram of techno-typological attributes analysed in this study.** The right column shows the final selection of attributes following the amalgamation of certain categories.
(PDF)

**S2 Fig. Heatmap of Jaccard distance values between lithic assemblages.** Values in the heatmap range from 0 (identical presence of techno-typological attributes) to 1 (complete absence of shared techno-typological attributes).
(PDF)

**S1 File. Raw data and corresponding R script.**
(ZIP)

**S2 File. Supplementary analyses and figures.**
(DOCX)

## Acknowledgments

The authors thank Alfredo Cortell-Nicolau, Eduardo de la Peña and Carlos Rodríguez-Rellán for discussions on spatial autocorrelation and Mantel tests. Open access funding was provided by the University of Vienna, which we also gratefully acknowledge.

## Author contributions

**Conceptualization:** Timothy Canessa, Paloma de la Peña.

**Data curation:** Timothy Canessa.

**Formal analysis:** Timothy Canessa.

**Funding acquisition:** Timothy Canessa.

**Investigation:** Timothy Canessa.

**Methodology:** Timothy Canessa, Paloma de la Peña.

**Visualization:** Timothy Canessa.

**Writing – original draft:** Timothy Canessa.

**Writing – review & editing:** Timothy Canessa, Paloma de la Peña.

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
