## [Decision Letter · Decision Letter 0]

29 Dec 2025

PONE-D-25-59950
Unpacking lithic assemblage variability in the Early Upper Palaeolithic: A multivariate approach to the structure of the Iberian Aurignacian
PLOS One

Dear Dr. Canessa,

Thank you for submitting your manuscript to PLOS ONE. After careful consideration, we feel that it has merit but does not fully meet PLOS ONE’s publication criteria as it currently stands. Although reviewers’ reports regard certain issues about the methods, the quality of the data and the interpretation of results, all of them express positive evaluations about the goal of your work, which is a unique opportunity to compare Aurignacian assemblages across space and time in Iberia. Main reviewers’ remarks turn around the sampling criteria, the data structure and composition, the data analysis (CA), all of them so fundamental to support the hypothesis and final interpretations of the author, alongside the use of terminology. Constructive reccomendations are reported alongside minor concerns and some references. Therefore, we invite you to submit a revised version of the manuscript that addresses the points raised during the review process.

We look forward to receiving your revised manuscript.

Kind regards,

Marco Peresani

Academic Editor

PLOS One

Journal Requirements:

2. In your manuscript, please provide additional information regarding the specimens used in your study. Ensure that you have reported human remain specimen numbers and complete repository information, including museum name and geographic location.

For more information on PLOS One's requirements for paleontology and archeology research, see https://journals.plos.org/plosone/s/submission-guidelines#loc-paleontology-and-archaeology-research.

4. We note that Figure 1 in your submission contain map images which may be copyrighted. All PLOS content is published under the Creative Commons Attribution License (CC BY 4.0), which means that the manuscript, images, and Supporting Information files will be freely available online, and any third party is permitted to access, download, copy, distribute, and use these materials in any way, even commercially, with proper attribution. For these reasons, we cannot publish previously copyrighted maps or satellite images created using proprietary data, such as Google software (Google Maps, Street View, and Earth). For more information, see our copyright guidelines: http://journals.plos.org/plosone/s/licenses-and-copyright.

Reviewers' comments:

Reviewer's Responses to Questions

**Comments to the Author**

1. Is the manuscript technically sound, and do the data support the conclusions?

Reviewer #1: Partly

Reviewer #2: Yes

Reviewer #3: Yes

2. Has the statistical analysis been performed appropriately and rigorously? 

Reviewer #1: N/A

Reviewer #2: Yes

Reviewer #3: Yes

3. Have the authors made all data underlying the findings in their manuscript fully available?

Reviewer #1: Yes

Reviewer #2: Yes

Reviewer #3: Yes

4. Is the manuscript presented in an intelligible fashion and written in standard English?

Reviewer #1: Yes

Reviewer #2: Yes

Reviewer #3: Yes

5. Review Comments to the Author

Reviewer #1: Review

The article “Unpacking lithic assemblage variability in the Early Upper Palaeolithic: A multivariate approach to the structure of the Iberian Aurignacian” authored by T. Canessa and P. de la Peña, presents a quantitative approach based on techno-typological data from Aurignacian sites in Iberia. The authors applied multivariate statistics to lithic datasets to characterise the spatial and temporal structure of the Aurignacian in Iberia. This work aims at identifying if there are inter-assemblage variability, both diachronic and synchronically among Aurignacian sites in Iberia, by applying a previously non-explored quantitative approach. The article introduces in detail the topic of study, first with the Aurignacian itself and then with the Early Upper Palaeolithic in Iberia, exhibiting extensive knowledge of the topic. The figures are pertinent, helping to understand the text and clarifying the main aspects of the manuscript. The text is well written, without typos, and with a good exposition of the data. The authors provide an Rscript and all the rawdata for the replicability of the study, aligning with Open Science practices which are well valuated in research community. All this makes it an important contribution to the field of archaeological research. However, there are certain issues that concerns me regarding the methods, the quality of the data and the interpretation of the results.

I therefore consider these changes, especially those concerning the data composition and the data analysis (CA), which are fundamental to support the hypothesis and final interpretations of the author, must be addressed.

I will explain my main concerns here followed by some minor comments at the end of the review.

Introduction

I would rework the introduction ensuring some clarity. It starts with the IUP, although there are not much information about it, then it could start directly from the Aurignacian (AU since now in advance) as it is the topic of this work. The explanation of the AU (Line 52) should have some of the references mentioned below (Line 58). But more importantly, the authors should note that the fours subdivisions of the AU are already explained by G.Laplace (1966) and then it should be the main reference used. Why it is only used the framework of Aquitaine and not the one from the Italian Peninsula, or the core sequence of Central Europe (Swabian Jura, Moravia..)

Also, as a suggestion, perhaps the authors could be more selective with literature to avoid overciting (81-83; 97-101).

Line 111: “how do we reconcile this variability with the notion that the Aurignacian is a shared tradition? “ This question is ambiguous to me, as variability has always existed in lithic assemblages and it is recognised within cultures. Another question is, what is and atypical assemblage? Atypical with respect to what? (Line 112). In my opinion, this term is rooted in a deep typological perspective, in which lithic assemblages were explained based on diagnostic artefacts. I recommend to reconsider using this terminology. If the authors aim at identifying variability among assemblages, then, atypical should not be used.

Methods

The authors explained they used 41 datasets from the Early Upper Palaeolithic but they should mention which were the criteria used for the selection of data. Do they collect all the available datasets in the literature? Were some sites excluded? If so, what was the criteria used to not include those sites? The authors should detail these aspects.

In parallel to this, another major concern is the quality of the data collected. Specially in the election of the sites or the comparable attributes.

Concerning the sites, for example, level 18b and 18c of El Castillo has been already attributed to Neanderthals (Garralda et al. 2022) because of the presence of three Neanderthal deciduous teeth, excluding the possibility of anatomically modern humans as the makers of this assemblage and so, its Aurignacian adscription. Level CHm.5 of Gorham’s Cave reported 17 artefacts in a disputable context. Same occurs in Aitzbitarte Vb where Aurignacian-Mousterian mixing was identified in the upper layer. Level XIIIinf of La Viña has been affected by post-depositional processes resulting in mix with Middle Palaeolithic units (Santamaria, 2012). Pego do Diabo is another controversial site. Excavated by spits, its level 2 lies between the Gravettian and Aurignacian attribution, but no clear evidence of the last one. One might wonder to what extent these examples can distort the subsequent analysis. In mi opinion, level 18b and 18c from El Castillo must be removed.

The authors could provide alternative analysis with a better-chosen sample of sites, or at least after removing the ones highly controversial for its chronocultural attribution (e.g. Castillo, Morin, Gorham’s, Pego do Diabo, etc.). This new analysis could be cross compared to see if the final results excluding controversial sites match with the ones they have already obtained.

Also, the authors should clarify the distinction they do about north and south. And why they preferred to use this distinction among others base on the biogeographical space (Vidal-Cordasco et al. 2022)

I would like to know if the authors have considered possible discrepancies with some dates of the assemblages that has low agreements in order to calculate the median. And to what extent the temporal variable can be the result of different lab methods or treatments.

Concerning the attributes, the application of multivariate statistics is good for make the data homogeneous. However, I am concerned about which data is included in the analysis and if these choices are contributing to biased results. First, the attributes are more typological than technological, and it can be seen in the Discussion, where most of the insights are based on the presence of retouched tools from a typological perspective, which is applying the same perspective the authors are trying to overcome/solve. Second, I wonder how comparable assemblages over 5000 artefacts, 1000 artefacts or less than 100 artefacts can be? Do they have the same relevance? From my perspective, 1 Dufour bladelet (which already means presence) over 5000 artefacts should not represent the same than 1 Dufour bladelet over 60 artefacts. And here it can be also discussed different occupation models and interpretations of the assemblages. Third, fossil directors are always reported in techno-typological studies, but not all the variables the authors gathered. Even though, they tried to concentrate different name conventions within a single term appropriately. I have these concerns with bipolar technology (addressed by the authors but included in the analysis), flake cores (which kind of reduction?). Also, flake core, burin core and carinated cores are not gathering the variability of reduction strategies in the EUP or AU and it is a weakness in the analysis. And last, did the authors consider the data that were missing in the original datasets? Did they account them as “not present”? This might revealed different results.

If the authors applied any strategy to mitigate inter-analyst biased or to increase lithic replicability, they should also report it in the method section (Pargeter et al. 2023).

Finally, the methods are considerably long and I suggest the authors condense the attributes presented in this section in the main text and set the details in a supplementary document. This will increase readability of the main points of the article and give place to explain what a Correspondence Analysis is (no need to explain it in the article as it is a worldwide term used in research)

Garralda, MD., Maíllo-Fernández, JM., Maureille, B. et al.  > 42 ka human teeth from El Castillo Cave (Cantabria, Spain) Mid-Upper Paleolithic transition. Archaeol Anthropol Sci 14, 126 (2022). https://doi.org/10.1007/s12520-022-01587-2

Pargeter J, Brooks A, Douze K, et al. Replicability in Lithic Analysis. American Antiquity. 2023;88(2):163-186. https://doi.org/10.1017/aaq.2023.4

Vidal-Cordasco, M., Ocio, D., Hickler, T. et al. Ecosystem productivity affected the spatiotemporal disappearance of Neanderthals in Iberia. Nat Ecol Evol 6, 1644–1657 (2022). https://doi.org/10.1038/s41559-022-01861-5

Results

The authors reported very interesting results suggesting that Aurignacian assemblages in Iberia are highly variable and those differences can be tracked spatially but not diachronically. One special mention is to the identification of some discrepancies in the chronological order of chronocultural markers as Dufour subtype Dufour bladelets and Dufour subtype Roc de Combe bladelets. They expand these ideas in the discussion, also mentioning that probably, the reference we used for structuring the Aurignacian are not that consistent than previously thought.

However, although these results are interesting and deserve to be further explored, I am not sure how reliable the author interpretations are looking at the CA analysis. The sum of the two dimensions account to 24%. This is a limited representation of the total contributions of the variables. Although the Eigenvalues are quite low in the 3, 4, 5, 6 dimensions, they could also represent alternative CA to see if the trends identified can be also supported by a more complete analysis of the data.

The authors should provide CA of the other dimensions (e.g. 3 and 4; 5 and 6; 3 and 5, 1 and 3) to make their interpretations stronger. This is relevant as trends observed the first and second dimensions can differ from other identified in the third, the fourth or the fifth.

Another solution is reducing the variables included in the analysis, especially the ones that clearly differs from the Aurignacian attribution and coincide with sites that has reported stratigraphic inconsistencies (e.g Chatelperronian points, MP points, LevalloisTech, DiscoidalTech, Microgravette). This will clean the data for the “noise” created by reporting lithic assemblage with possible artefact mixing or inconsistent contextual data.

The authors must conduct these steps to increase the reliability of the analysis and make their assumptions (Line 722-727 among others) more robust. I encourage them to dedicate time on this regard in order to build consistent quantitative analysis for the arrival of AMH into Iberia.

Discussion

Same mention for the Introduction section. I will avoid the term atypical or non-diagnostic from the discussions as they refer to what it should be included or not in an assemblage. But variability is part of the palaeolithic world, different populations organising technology in so many diverse ways according to factors such as duration, intensity, mobility, raw material availability, skills, culture. There are so many factors that can be behind the variability, but assemblages lacking chronocultural markers should not be defined as atypical, as there are not typical assemblages in palaeolithic archaeology. For example, the presence of several Chatelperronian points in small assemblages in Labeko Koba, Ekain or Cova Foradada have been interpreted as logistic hunting camps (Rios-Garaizar et al. 2012 and Morales et al. 2019). Chatelperronian points would make typical this assemblage in the Chatelperronian phylum? However, the limited number of artefacts make unable any information about average technological traits, reduction sequences, management of raw materials, refittings, spatial organisation, etc. Assemblages with no fossil-directors perhaps suggests that those ones are not that relevant to understand or study human behaviour and cultural phases among the Upper Palaeolithic record.

This is partially explained in Lines 811-813, but those assemblages are incorporated to the data as part of the “variability” of the EUP in Iberia with any additional explanation or reflection about what do they mean with regards the general structure of the Aurignacian. The authors recognise this limitation (line 865-868)

Conclusions

Also, at the end the conclusions are not very well linked with the Introduction and some of the questions mentioned in that section. The authors might explore whether the spatial similarities among assemblages are also consistent with the interpretative scenarios mentioned for dispersal of the AMH population in Iberia or previous models discussed such as the Ebro Frontier.

Morales, J. I., A. Cebrià, A. Burguet-Coca, J. L. Fernández-Marchena, G. García-Argudo, A. Rodríguez-Hidalgo, M. Soto, S. Talamo, J.-M. Tejero, J. Vallverdú and J. M. Fullola (2019). "The Middle-to-Upper Paleolithic transition occupations from Cova Foradada (Calafell, NE Iberia)." PLOS ONE 14(5): e0215832. https://doi.org/10.1371/journal.pone.0215832

Rios-Garaizar, J., A. Arrizabalaga and A. Villaluenga (2012). "Haltes de chasse du Châtelperronien de la Péninsule Ibérique : Labeko Koba et Ekain (Pays Basque Péninsulaire)." L'Anthropologie 116(4): 532-549. https://doi.org/10.1016/j.anthro.2012.10.001

Minor comments

53-54 Repetition: novelty and innovations.

55 and 1019 tool-making groups is not a common term, use better “human groups”, “hunter gatherer groups”

311 – Cova Gran is a EUP assemblage with no Middle Palaeolithic component.

711 I found this idea oversimplified. Of course there are networks among hunter-gatherers

717 I suggest removing the idea of exchange as it is not proven and long-distance raw materials are a primary indicator of mobility.

758 Correction: The use of poor-quality chalcedony does not apply to the Magdalenian sequence of Cova Gran. There are recent works addressing this, also in Cova del Parco cave, also Magdalenian, that incorporates good quality evaporitic chert in the lithic assemblages. For some raw material constraints in the technological organisation, sites as La Viña, Pego do Diabo or Gato Preto reported industries mainly based on quartz.

Line 642 and Line 773 (among others) I will avoid the use of the term “atypical” or “non-diagnostic” of the discussion.

All in all, I believe this is a soundness contribution to the field that I hope the authors improve taking into account the suggestions here mentioned.

Reviewer #2: The paper by Canessa and de la Peña addresses one of the most intriguing regions in EUP research: the Iberian Peninsula. The archaeological record here seems to differ significantly from the broader models of change visible in France and Italy, particularly regarding the southern record. As the authors state, the variability in Iberia does not easily fit into these established European frameworks, a factor that contributed to J. Zilhão’s model of a very late Aurignacian presence in southern Iberia. While the picture is slowly changing thanks to new excavations and dating programs, certain issues remain open to debate (e.g., Bajondillo). In this context, this quantitative study is a welcome contribution.

First, I want to state that the goal of the authors is remarkable; this represents one of the few attempts to compare Aurignacian assemblages quantitatively across space and time. I truly appreciate the open sharing of datasets and the methodological workflow. The supplementary information is clear and well-structured, and I assume it will be published as a repository with a DOI based on the data statement. I was able to reproduce the main findings without issue using the well-written and organized R code. I also commend the additions to the Mantel tests to discuss spatial autocorrelation, as well as the use of an R package to calibrate dates and extract median ages. This aspect of the work is executed exceptionally well. However, despite these strengths, I have several concerns regarding the underlying data structure and methodology that need to be addressed before publication:

Presence/absence data: My main skepticism lies in the use of presence/absence data for testing variability across the Aurignacian. I question whether this approach is sufficiently high-resolution for the Upper Paleolithic, and specifically for a technocomplex like the Aurignacian, which is often defined by a set of features (e.g., carinated cores, Dufour bladelets) that are ubiquitous across its temporal span. The risk here is oversimplification, which may mask variability. Recent studies on the Aurignacian demonstrate that it is often the frequency or relative importance of specific technological features, rather than their presence, that discriminates between different phases. By relying on presence/absence, the discussion risks becoming blurred. While Blinkhorn and Grove used a similar approach, they analyzed a much larger temporal and geographic span with MSA sites structured by raw materials and variables such as roughness. However, their scope and research questions were fundamentally different. I appreciate that the authors discuss these shortcomings in the discussion section, but I believe further validation is required.

Raw material variability: One way to strengthen the analysis is to test more directly if raw material availability affects the observed variability. Although I understand that data availability is limited, the strong spatial signal observed suggests that raw material constraints may play a significant role. Could the authors attempt to use broad groups of raw materials to test the outcomes? I suspect, for instance, that the presence of Quartz drives significant variability. While raw material cannot explain all variability, it likely plays a crucial role that should be explicitly tested, even if only at a coarse resolution.

Regional sensitivity: I propose running an additional analysis focused exclusively on northern Iberia, where the data is more fine-grained and the dating is generally more reliable. Since the authors noted that northern assemblages appear more similar to one another, it would be valuable to see if the Mantel tests and partial Mantel tests yield different results when restricted to this region.

Stratigraphic integrity and typological noise I argue that a presence/absence analysis must account for stratigraphic mixing and excavation errors, particularly when dealing with this specific set of sites. What happens if the analysis focuses on types clearly linked to the Aurignacian, excluding elements like Châtelperronian points? The inclusion of such types, or Levallois technology (which is not clearly documented in the Aurignacian), suggests that the analysis may be picking up on stratigraphic or post-depositional mixing rather than cultural variability. Looking at Figures 5 and 6, it appears that some of these "intrusive" blanks drive a large amount of variability in the CA analysis. Also, categories such as "core working" are not particularly useful in this context, as they are too broad to provide meaningful distinctions.

In conclusion, I truly appreciate the competent and mature discussion provided in the manuscript. I believe this paper will serve as a starting point for expanding this specific research agenda. However, the authors need to more carefully address the limitations of presence/absence data and the necessity of collecting technological, quantitative data across sites to ensure the robustness of their conclusions.

A minor comment: Why are only partial Mantel tests reported in the text and table? It would be helpful to include the full results.

Reviewer #3: We would like first to thank the editors for entrusting us with the peer review of this study. We should note, however, that our expertise does not encompass multivariate analysis, therefore we did not comment the specific methodological details of the statistical approach. This review is divided into three parts: an introductory section, a series of general suggestions, and finally detailed line-by-line comments.

The authors present a large-scale statistical analysis of lithic typo-technological data from Early Upper Palaeolitic (EUP) assemblages spanning about 10,000 years (43-32 ka cal BP) across the Iberian peninsula. The analysis aims to unravel the variability within and between the chrono-cultural technocomplexes characteristic of this period, namely the Proto-Aurignacian, Early Aurignacian, Evolved Aurignacian, and Late Aurignacian. Some assemblages lack diagnostic features typical of Aurignacian technocomplexes, raising the question of the validity and relevance of using “index fossils” to define archaeological cultures in this region and timeframe. To address this issue, the authors perform a combination of two multivariate analytical methods to test the correlation between inter-assemblage variability and spatial and temporal distances. Data were collected from the archaeological literature and operationalised as binary variables (presence/absence of traits), enabling the inclusion of an extensive number of assemblages (n=41). The dataset comprises both technological (n=13) and typological attributes (n=18).

The authors demonstrate that spatial distance significantly influences inter-assemblage lithic typo-technological variability. They highlight a marked divide between northern and southern assemblages, arguing that spatial proximity is a strong predictor of similarity. While several explanatory factors are considered (including information networks, foraging strategies, raw material availability, and taphonomic biases), the authors propose that inter-assemblage variability is mainly driven by resource distribution, and associated mobility patterns and social organisation. In parallel, they show that temporal distance does not affect inter-assemblage lithic typo-technological variability, thereby challenging the conventional chronological succession of the four Aurignacian phases as inferred from southwestern French archaeological sequences (the so-called “Aquitaine model”). The observed inconsistency of diagnostic Aurignacian features (e.g. carinated cores, Dufour bladelet sub-types, Aurignacian retouch), combined with the occurrence across technocomplexes of several undiagnostic traits (e.g. bipolar, discoid and ubiquitous flaking methods) lead the authors to questioning the spatio-temporal validity of these “index fossils.” They ultimately advocate shifting the focus from the characterisation of archaeological cultures, potentially shaped by arbitrary research biases, towards a more process-oriented understanding of technological and behavioural change across this period.

In our view, this study offers an original and much-needed approach to the investigation of the EUP. The extensive spatial and temporal scope (the Iberian peninsula over more than 10,000 years), along with the substantial number of assemblages (n=41) and typo-technological attributes (n=31), are particularly noteworthy. This work represents a highly convincing effort of quantification of carefully selected qualitative data, and stands as an important contribution to cumulative, heuristic research, particularly through its commitment to open science principles.

We would however like to suggest a few minor revisions, particularly in the discussion section.

General comments and suggestions

- Proportions of attributes:

The authors repeatedly acknowledge the limitations of simplifying lithic assemblage variables into binary presence/absence traits. We find their reasoning convincing and appreciate the transparency maintained throughout the paper. In particular, the argument that inconsistencies in the literature preclude the use of continuous variables (e.g. line 253) is well justified. However, we would suggest to further discuss the potential methodological biases arising from the lack of proportional representation of attributes. The binary system used in this study does not convey the relative importance of each attribute in relation with others. For instance, the proportion of core categories is often used to characterise technocomplexes. Excluding “generic core types” from the analysis (line 347) may therefore lead to an overestimation of the significance of diagnostic “index fossils” cores, such as carinated cores.

- Technological attribute:

Recent studies have suggested that bladelet core management through the extraction of lateral and often asymmetrical blades represents a characteristic technological feature of the Proto-Aurignacian (Falcucci et al. 2017; Falcucci and Peresani 2018; Gennai et al. 2021; Gennai et al. 2025). We suggest considering the inclusion of a technological attribute related to lateral core management via the removal of “comma-like” blades (Falcucci et al 2020) to test the validity of this criterion for identifying Proto-Aurignacian assemblages in the Iberian peninsula. We believe that including this attribute would effectively complement the current list, which is relatively oriented towards lithic typology (18 typological versus 13 technological attributes). We would however understand that, once again, the variety of the literature used may refrain from integrating such a technological category of artefacts into the list of attributes.

- Spatial representativity:

As said before, the large spatial scale of the study is greatly appreciated. Nevertheless, we would like to raise a few concerns regarding the spatial representativity of the sample, which, in our opinion, warrants further discussion. Since the spatial segmentation of chaînes opératoires is known to vary across technocomplexes (e.g. Bordes et al. 2006 ; Porraz et al. 2010), the archaeological interpretation of the absence of diagnostic pieces (e.g. lines 909-912 and 956-957) may be affected by the incomplete excavation of occupations. This issue may be particularly pronounced at raw material procurement sites, where the range of activities is often very limited, and may be spatially segmented across broader territories. Furthermore, among the assemblages (n=41), only two sites are open-air, both attributed to the Evolved/Late Aurignacian. This raises the possibility that the presence or absence of variables, and consequently the potentially changing “functional solutions” observed across spatial and temporal scales (line 943), may depend heavily on the spatial (and thereof chronological) distribution of archaeological research itself. We believe this limitation has not been sufficiently emphasised in the current version of the manuscript. Nevertheless, we recognise that the dataset derives from published sources and thus inevitably reflects diverse approaches and research traditions, which the authors have admirably synthesised, and standardised in a nuanced manner.

- Bipolar and discoid methods:

We find the discussion of undiagnostic attributes across spatial and temporal dimensions (lines 923-924) particularly engaging. However, while the authors address generic flaking methods, they only briefly mention bipolar and discoid methods, which, from our perspective, appear far less ubiquitous during the Upper Palaeolithic than generic flake production, and therefore potentially more informative. In particular, are bipolar and discoid methods used exclusively to produce flakes, or are they also integrated into laminar reduction sequences? We would encourage the authors to elaborate further on the significance and distribution of bipolar and discoid methods across EUP assemblages.

- Cross-regional perspective:

The Aquitaine model is primarily grounded in qualitative technological analyses of long archaeological sequences in southwestern France. To our knowledge, no large-scale quantitative approach comparable to the one proposed here has yet been undertaken in this region. We consider this absence a significant limitation for meaningful cumulative comparisons between southwestern France and the Iberian peninsula, thereby hindering any conclusive attempts to evaluate or challenge the Aquitaine model. Given that large-scale quantitative studies remain scarce, we recommend integrating in the manuscript what appears to be the only other comparable and recent effort to assess technological similarity among EUP assemblages, namely Gennai et al. 2025 (https://doi.org/10.1371/journal.pone.0331393). Although based on a smaller set of assemblages, spanning a narrower temporal range but a broader spatial scale, this dataset was prepared for multivariate analysis and is openly available, allowing it to be processed according to the methodology employed in the present study and directly compared with the Iberian results.

Looking ahead, we enthusiastically encourage the authors to broaden even more the spatial coverage of their analysis, potentially through collaboration with teams engaged in similar quantitative research. We strongly believe that fostering convergence among research groups working in different geographical areas under open science standards will be a crucial step towards a deeper understanding of the processes of technological and cultural change during the EUP.

- Aurignacian expressions:

We greatly appreciate the broader interpretations derived from the empirical results, particularly the proposition at lines 955-959 that inconsistent proportions of diagnostic and undiagnostic features in assemblages may alternatively reflect either a diverse manifestation of a generic and widely shared Aurignacian technological system, or differing strengths of ties between groups within this spatial and temporal frame, as expressed by the scattered emergence of “low-fidelity innovations.” Yet, whether these should be termed “innovations” is debatable, see for instance the introduction in O’Brien and Shennan (ed) for different definitions of the term.

Line-by-line comments and suggestions

- 72: “discreet” → “discrete”

- 96: add Gennai et al. 2025, where they assess the technological variability across EUP sites

- 150: add Djakovic et al. 2022, where they discuss the timing of overlap between CP and PA

- 341: “Carinated.EndsraperCores” → “Carinated.EndscraperCores”

- 404-405: add Le Brun-Ricalens et al. 2009, where they discuss the typological problem of Krems versus Font-Yves points

- 417-418: “piéces esquilles” → “pièces esquillées”

- 463: “Within in each” → “Within each”

- 509: “Aurignacian blades, which are only known from northern sites”: does this statement only targets the Iberian peninsula or is it broader?

- 685-686: add Le Brun-Ricalens et al. 2009; Tsanova et al. 2012

- 744 and 746: numbers “3” and “2” should be written in letters

- 791: “of with 268” → “of which 268”

- 877: “Rècent” → “Récent”

- 1044: “phdthesis” → “PhD thesis”

- 1059, 1062, 1188: “Arrizabalaga, À.”: accent on À not appearing at lines 1050 and 1054

- 1178: “paléolithique” → “Paléolithique”

- 1267: “paleolithique superieur” → “Paléolithique supérieur”

- 1645: “tronquèes” → “tronquées”

- 1646: “de France” → “française”

- 1683: “Península” → “Péninsule”

- 1683-1884: “Paleolithique moyen recent et paleolithique superieur” → “Paléolithique moyen récent et Paléolithique supérieur”

- 1746-1747: remove squares

- 1757: “contain” to remove

References used in the review

Bordes, J.-G., 2006. ‘News from the West: a reevaluation of the classical Aurignacian sequences of the Périgord.’ In: Bar-Yosef, O., Zilhão, J. (Eds.). Towards a Definition of the Aurignacian. Lisboa: Instituto Português de Arqueologia, 147–171.

Djakovic, I., Key, A., Soressi, M, 2022. ‘Optimal linear estimation models predict 1400–2900 years of overlap between Homo Sapiens and Neandertals prior to their Disappearance from France and Northern Spain.’ Scientific Reports 12 (1): 15000. https://doi.org/10.1038/s41598-022-19162-z.

Falcucci, A., Conard, N. J., Peresani, M., 2017. ‘A critical assessment of the

Protoaurignacian lithic technology at Fumane Cave and its implications for the

definition of the earliest Aurignacian.’ PLOS ONE, 12(12), e0189241.

https://doi.org/10.1371/journal.pone.0189241.

Falcucci, A., Conard, N. J., Peresani, M., 2020. ‘Breaking through the Aquitaine frame: A

re-evaluation on the significance of regional variants during the Aurignacian as seen

from a key record in southern Europe.’ Journal of anthropological sciences = Rivista

di antropologia: JASS, 98, 99–140. https://doi.org/10.4436/JASS.98021.

Falcucci, A., Peresani, M., 2018. ‘Protoaurignacian core reduction procedures: blade and bladelet technologies at Fumane Cave’. Lithic Technology 43 (2): 125–40. https://doi.org/10.1080/01977261.2018.1439681.

Gennai, J., Falcucci, A., Niochet, V., Peresani, M., Richter, J., Soressi, M., 2025. ‘Tracking the emergence of the Upper Palaeolithic in Western Asia and Europe: a multiple correspondence analysis of Protoaurignacian and Southern Ahmarian lithics.’ PLOS ONE 20 (9): e0331393. https://doi.org/10.1371/journal.pone.0331393.

Gennai, J., Peresani, M., Richter, J., 2021. ‘Blades, bladelets or blade(let)s? Investigating early upper Palaeolithic technology and taxonomical considerations.’ Quartär 68: 71-116.

Le Brun-Ricalens, F., Bordes, J.-G., Eizenberg, L., 2009. ‘A crossed-glance between southern European and Middle-Near Eastern early Upper Palaeolithic lithic technocomplexes. Existing models, new perspectives.’ In: Camps, M., Szmidt, C. C., eds. The Mediterranean from 50 000 to 25 000 BP: Turning Points and New Directions. Oxford: Oxbow Books, 11–33.

O’Brien, M. J., Shennan, S. (Ed.), 2010. Innovation in Cultural Systems: Contributions from Evolutionary Anthropology. Vienna Series in Theoretical Biology, MIT Press.

Porraz, G., Simon, P., Pasquini, A., 2010. ‘Identité technique et comportements économiques des groupes proto-aurignaciens à la grotte de l’Observatoire (principauté de Monaco)’. Gallia préhistoire 52 (1): 33–59. https://doi.org/10.3406/galip.2010.2470.

Tsanova, T., Zwyns, N., Eizenberg, L., Teyssandier, N., Le Brun-Ricalens, F., Otte, M., 2012. ‘Le plus petit dénominateur commun : réflexion sur la variabilité des ensembles lamellaires du Paléolithique supérieur ancien d’Eurasie. Un bilan autour des exemples de Kozarnika (Est des Balkans) et Yafteh (Zagros central).’ L’Anthropologie 116 (4): 469–509. https://doi.org/10.1016/j.anthro.2011.10.005.

6. PLOS authors have the option to publish the peer review history of their article (what does this mean?). If published, this will include your full peer review and any attached files.

Reviewer #1: No

Reviewer #2: No

Reviewer #3: No

---

## [Author Response · Author response to Decision Letter 1]

12 Feb 2026

[Please consult the 'Response to Reviewers' document to see our responses in blue text]

12th February 2026

We greatly appreciate the consideration of our Manuscript in PLOS ONE and the time invested by the three anonymous reviewers. We take note of the points raised in the opening paragraph of the decision letter that refer to the study’s ‘sampling criteria’, ‘data structure and composition’, ’data analysis (CA)’ and ‘use of terminology’. While we have addressed individual reviewer comments in a later section of this letter, we nonetheless inform you that these individual elements have been tackled in the following way:

Sampling criteria:

We have addressed comments about the presence-absence format of lithic data and provided additional text in the Manuscript that highlights this potential limitation. Moreover, we have given greater emphasis to the fact different patterns could be observed with analyses of continuous variables (e.g., proportion of core types or bladelets) rather than the categorical variables we use in our study. We have also run additional analyses with different sample selection criteria for both techno-typological attributes and radiocarbon dates. These have been referenced in the Manuscript and presented separately in a new S2 File.

 In all cases these additional analyses with different criteria replicate the original results of the Manuscript.

Data structure and composition:

In line with the above, we have addressed concerns that the data may be too typologically driven and provided the results of further analyses with less typological attributes in the S2 File. Specific comments on the significance/insignificance of certain techno-typological attributes have also been addressed via individual responses.

Data analysis (CA):

Overlapping with the above, the results of a CA without potentially intrusive or characteristically non-Aurignacian attributes (i.e., LevalloisTech, ChâtelperronianPoints, MP.Points and Micogravettes) are mentioned in the Manuscript to support the original conclusions. The plots are additionally provided in the S2 File along with some text summarising the analyses.

Use of terminology:

Terminological issues raised by Reviewers 1 and 3 have been addressed, the details of which can be found in our responses. We have also refined the use of other terms not specifically highlighted.

Other:

We noticed a labelling error in Fig 1 and have now submitted a corrected version. Other typos or mistakes recognised during our rereading of the Manuscript have also been amended.

With regards to the Journal Requirements listed in the letter, we hereby confirm that each requirement has been adhered to in accordance with the following:

‘1.Please ensure that your Manuscript meets PLOS ONE's style requirements, including those for file naming.’

We confirm that the Manuscript meets all of PLOS ONE’s style requirements. As per Editorial Office’s request, we have amended the referencing style.

‘2. In your Manuscript, please provide additional information regarding the specimens used in your study. Ensure that you have reported human remain specimen numbers and complete repository information, including museum name and geographic location.’

No specimens were used in our study.

‘3. When completing the data availability statement of the submission form, you indicated that you will make your data available on acceptance. We strongly recommend all authors decide on a data sharing plan before acceptance, as the process can be lengthy and hold up publication timelines. Please note that, though access restrictions are acceptable now, your entire data will need to be made freely accessible if your Manuscript is accepted for publication. This policy applies to all data except where public deposition would breach compliance with the protocol approved by your research ethics board. If you are unable to adhere to our open data policy, please kindly revise your statement to explain your reasoning and we will seek the editor's input on an exemption. Please be assured that, once you have provided your new statement, the assessment of your exemption will not hold up the peer review process.’

We have created a Zenodo entry and attached its doi to the data availability statement. This doi link will be activated if/once the manuscript is accepted for publication, as we do not want our data to be used before the associated study is actually published. This follows the offer stated above, which permits restricted access until the Manuscript is accepted for publication.

‘4. We note that Figure 1 in your submission contain map images which may be copyrighted. All PLOS content is published under the Creative Commons Attribution License (CC BY 4.0), which means that the Manuscript, images, and Supporting Information files will be freely available online, and any third party is permitted to access, download, copy, distribute, and use these materials in any way, even commercially, with proper attribution. For these reasons, we cannot publish previously copyrighted maps or satellite images created using proprietary data, such as Google software (Google Maps, Street View, and Earth). For more information, see our copyright guidelines: http://journals.plos.org/plosone/s/licenses-and-copyright.’

We have amended Figure 1 following guidance from the Editorial Office. The Digital Elevation Model is now taken from the free to use USGS EROS Centre (https://www.usgs.gov/centers/eros/data). Use of the ocean bathymetry data (from the EMODnet) remains unchanged as it is licensed under Creative Commons CC-BY 4.0 (see https://emodnet.ec.europa.eu/en/terms-use-emodnet-online-services-data-and-data-products). The relevant figure caption has also been amended.

‘5. If the reviewer comments include a recommendation to cite specific previously published works, please review and evaluate these publications to determine whether they are relevant and should be cited. There is no requirement to cite these works unless the editor has indicated otherwise.’

We confirm that suggestions to cite specific published works have been reviewed and evaluated with accuracy.

With this said, we would like to thank the three anonymous reviewers for their constructive and insightful comments which have helped improve the Manuscript. We are especially grateful for the positive evaluation of our work and identification of certain weaknesses. Below we address each comment with a response in blue text and refer to line numbers from the revised Manuscript. Please note that the line numbers in Microsoft Word been observed to shift after saving the document so the numbers provided here may be marginally incorrect.

Reviewer #1: Review The article “Unpacking lithic assemblage variability in the Early Upper Palaeolithic: A multivariate approach to the structure of the Iberian Aurignacian” authored by T. Canessa and P. de la Peña, presents a quantitative approach based on techno-typological data from Aurignacian sites in Iberia. The authors applied multivariate statistics to lithic datasets to characterise the spatial and temporal structure of the Aurignacian in Iberia. This work aims at identifying if there are inter-assemblage variability, both diachronic and synchronically among Aurignacian sites in Iberia, by applying a previously non-explored quantitative approach. The article introduces in detail the topic of study, first with the Aurignacian itself and then with the Early Upper Palaeolithic in Iberia, exhibiting extensive knowledge of the topic. The figures are pertinent, helping to understand the text and clarifying the main aspects of the Manuscript. The text is well written, without typos, and with a good exposition of the data. The authors provide an Rscript and all the rawdata for the replicability of the study, aligning with Open Science practices which are well valuated in research community. All this makes it an important contribution to the field of archaeological research. However, there are certain issues that concerns me regarding the methods, the quality of the data and the interpretation of the results. I therefore consider these changes, especially those concerning the data composition and the data analysis (CA), which are fundamental to support the hypothesis and final interpretations of the author, must be addressed. I will explain my main concerns here followed by some minor comments at the end of the review. 

We appreciate the recognition of our work and its contribution to open science. We have addressed the issues raised regarding its methods, the quality of the data and interpretation below under individual comments.

Introduction

I would rework the introduction ensuring some clarity. It starts with the IUP, although there are not much information about it, then it could start directly from the Aurignacian (AU since now in advance) as it is the topic of this work. The explanation of the AU (Line 52) should have some of the references mentioned below (Line 58). But more importantly, the authors should note that the fours subdivisions of the AU are already explained by G.Laplace (1966) and then it should be the main reference used. Why it is only used the framework of Aquitaine and not the one from the Italian Peninsula, or the core sequence of Central Europe (Swabian Jura, Moravia..) Also, as a suggestion, perhaps the authors could be more selective with literature to avoid overciting (81-83; 97-101).

 Our brief mention of IUP assemblages serves to provide context to the Aurignacian of the subsequent EUP. This is because, in contrast to the circumscript nature of IUP assemblages (not all of which are currently and exclusively associated with Homo sapiens), the Aurignacian is a technocomplex with extensive geographic and chronological distribution. We therefore aimed to emphasise that this is one of the qualities which renders it a topic of enduring interest in Palaeolithic archaeology. To this end, we have reworked the opening paragraph to make this point clearer.

 With regards to work of G. Laplace (1966), he divided the Aurignacian into four main phases (Protoaurignacien, Aurignacien Ancien, Aurignacien Évolué and Aurignacien Évolué Final) and suggested further subdivisions for most of these based on their typological compositions e.g. Aurignacien Évolué à fiable indice de burins and Aurignacien Évolué à fort indice de burins. While we acknowledge his work as an important contribution, we are inclined to argue that it is not the main reference for the Iberian Aurignacian. The four subdivisions we list in line 68 are the naming conventions mostly commonly used in the Iberian literature. These are collectively based on the the many works we cite in the paragraph and recently refined with archaeological and chronometric evidence. To make this clearer, we have added the word ‘consolidation’.

 The Aquitaine framework is referred to as the leading framework because is the one that has traditionally informed upon the identification and characterisation of the Iberian Aurignacian (e.g. Zilhão 1997, Villaverde et al. 1998), not least because of the proximity to SW France and the influence of French scholarship on Palaeolithic research in Iberia. Therefore, for our study and its questions, we believe that the frameworks derived from Italy or Central Europe do not supersede the Aquitaine one.

 We accept the suggestion of being more selective with citations and have done so. We have also added Gennai et al. 2025 following Reviewer 3’s suggestion.

Line 111: “how do we reconcile this variability with the notion that the Aurignacian is a shared tradition? “ This question is ambiguous to me, as variability has always existed in lithic assemblages and it is recognised within cultures. Another question is, what is and atypical assemblage? Atypical with respect to what? (Line 112). In my opinion, this term is rooted in a deep typological perspective, in which lithic assemblages were explained based on diagnostic artefacts. I recommend to reconsider using this terminology. If the authors aim at identifying variability among assemblages, then, atypical should not be used.

We agree that variability has commonly been observed within lithic industries, even when some influential scholars have emphasised a more homogenous, pan-European picture of industries. What has been less explored, however, is whether the extent of this variability lends support to the idea that the Iberian Aurignacian is a strongly represented and shared tradition (a point we later address in the Discussion). We have now modified the wording to make this clearer.

 We also welcome the question ‘what is an atypical assemblage?’ as a valid one. In our question posed in the Introduction, we are referring to assemblages which are atypical in the cultural taxonomic sense. Of course, from a behavioural perspective ‘atypical’ is misleading. We have hence replaced the term ‘atypical’ throughout the paper.

Methods The authors explained they used 41 datasets from the Early Upper Palaeolithic but they should mention which were the criteria used for the selection of data. Do they collect all the available datasets in the literature? Were some sites excluded? If so, what was the criteria used to not include those sites? The authors should detail these aspects.

We appreciate if the criteria of assemblage selection was not fully clear based on the information provided in lines 247–284. For the avoidance of doubt, we aimed to include as many assemblages as possible that are archaeologically and/or chronologically relevant to the Aurignacian in Iberia; in other words, assemblages claimed to be Aurignacian based on their lithic artefact component or assemblages dated to the Aurignacian timeframe but which have been classified differently (e.g. ‘indeterminate Early Upper Palaeolithic’) or not at all (e.g. Lapa do Picareiro). In connection, as our analysis is concerned with both the technological and typological composition of these assemblages, only assemblages with sufficient published data were selected (it is for this reason that the Abstract states “all sufficiently published and chronologically relevant assemblages” ). This means that, for example, assemblages with only retouched tool counts published could be not included in the analysis for lack of technological information (for publications from the 20th century, this is unfortunately a common occurrence).

 In parallel to this, another major concern is the quality of the data collected. Specially in the election of the sites or the comparable attributes. Concerning the sites, for example, level 18b and 18c of El Castillo has been already attributed to Neanderthals (Garralda et al. 2022) because of the presence of three Neanderthal deciduous teeth, excluding the possibility of anatomically modern humans as the makers of this assemblage and so, its Aurignacian adscription. Level CHm.5 of Gorham’s Cave reported 17 artefacts in a disputable context. Same occurs in Aitzbitarte Vb where Aurignacian-Mousterian mixing was identified in the upper layer. Level XIIIinf of La Viña has been affected by post-depositional processes resulting in mix with Middle Palaeolithic units (Santamaria, 2012). Pego do Diabo is another controversial site. Excavated by spits, its level 2 lies between the Gravettian and Aurignacian attribution, but no clear evidence of the last one. One might wonder to what extent these examples can distort the subsequent analysis. In mi opinion, level 18b and 18c from El Castillo must be removed.

 Respectfully, we do not agree with the idea that all archaeological materials of levels 18B and 18C in El Castillo can be exclusively attributed to Neanderthals based on metric and qualitative attributes (rather than genetic or proteomic evidence, or the morphology of the internal structures of the teeth) of three worn deciduous tooth crowns. Indeed, to exclude the possibility of an anatomically modern human component would assert that the Neanderthals were exclusively responsible for the characteristic Aurignacian (e.g. bone tools, carinated- and nosed-endscraper cores, Aurignacian blades and Dufour bladelets [Fig. 2]) and general Upper Palaeolithic elements like blades, bladelets and end-scrapers.

---

## [Decision Letter · Decision Letter 1]

3 Mar 2026

Unpacking lithic assemblage variability in the Early Upper Palaeolithic: A multivariate approach to the structure of the Iberian Aurignacian

PONE-D-25-59950R1

Dear Dr. Canessa,

We’re pleased to inform you that your manuscript has been judged scientifically suitable for publication will be formally accepted for publication once it meets all outstanding technical requirements. Although two reviewers noted that your manuscript took advantage of only partial improvements despite some interesting suggestions raised, they do not believe you will address further points in a second round of revision. I converge with both of them in thinking that your work will contribute to renew reasonements on the Aurignacian in Iberia.

Kind regards,

Marco Peresani

Academic Editor

PLOS One

Additional Editor Comments (optional):

Reviewers' comments:

Reviewer's Responses to Questions

**Comments to the Author**

1. If the authors have adequately addressed your comments raised in a previous round of review and you feel that this manuscript is now acceptable for publication, you may indicate that here to bypass the “Comments to the Author” section, enter your conflict of interest statement in the “Confidential to Editor” section, and submit your "Accept" recommendation.

Reviewer #1: All comments have been addressed

Reviewer #2: (No Response)

Reviewer #3: All comments have been addressed

2. Is the manuscript technically sound, and do the data support the conclusions?

Reviewer #1: Yes

Reviewer #2: Yes

Reviewer #3: Yes

3. Has the statistical analysis been performed appropriately and rigorously? 

Reviewer #1: Yes

Reviewer #2: Yes

Reviewer #3: Yes

4. Have the authors made all data underlying the findings in their manuscript fully available?

Reviewer #1: Yes

Reviewer #2: Yes

Reviewer #3: Yes

5. Is the manuscript presented in an intelligible fashion and written in standard English?

Reviewer #1: Yes

Reviewer #2: Yes

Reviewer #3: Yes

6. Review Comments to the Author

Reviewer #1: (No Response)

Reviewer #2: Overall, while a few parts have been improved, the authors did not truly engage with the level of revision requested by all reviewers. I understand that addressing all points would have been more time-consuming, but there were some very interesting suggestions raised. In particular, I would have liked to see the analysis conducted on a subset of the data (northern Iberia. Is it because a pattern can be seen there?). That being said, I do not believe the authors will address further points in a second round of revision. Since the work holds value, I suggest accepting it as it stands to spark a renewed and much-needed discussion on the Aurignacian in Iberia.

Reviewer #3: All my previous comments were addressed thoroughly. I have no further additions nor corrections to suggest.

7. PLOS authors have the option to publish the peer review history of their article (what does this mean?). If published, this will include your full peer review and any attached files.

Reviewer #1: No

Reviewer #2: No

Reviewer #3: No

---

## [Editor Report · Acceptance letter]

PONE-D-25-59950R1

PLOS One

Dear Dr. Canessa,

I'm pleased to inform you that your manuscript has been deemed suitable for publication in PLOS One. Congratulations! Your manuscript is now being handed over to our production team.

Kind regards,

on behalf of

Dr. Marco Peresani

Academic Editor

PLOS One